# Spermidine-mediated hypusination of translation factor EIF5A improves mitochondrial fatty acid oxidation and prevents non-alcoholic steatohepatitis progression

Jin Zhou [1,8] ✉, Jeremy Pang[1,8], Madhulika Tripathi[1], Jia Pei Ho[1], Anissa Anindya Widjaja [1], Shamini Guna Shekeran[1], Stuart Alexander Cook [1,2,3], Ayako Suzuki[4], Anna Mae Diehl [4], Enrico Petretto [1,2,5], Brijesh Kumar Singh[1] & Paul Michael Yen [1,6,7] ✉

Spermidine is a natural polyamine that has health benefits and extends life span in several species. Deoxyhypusine synthase (DHPS) and deoxyhypusine hydroxylase (DOHH) are key enzymes that utilize spermidine to catalyze the post-translational hypusination of the translation factor EIF5A (EIF5A[H]). Here, we have found that hepatic *DOHH* mRNA expression is decreased in patients and mice with non-alcoholic steatohepatitis (NASH), and hepatic cells treated with fatty acids. The mouse and cell culture models of NASH have concomitant decreases in Eif5a[H] and mitochondrial protein synthesis which leads to lower mitochondrial activity and fatty acid β-oxidation. Spermidine treatment restores EIF5A[H], partially restores protein synthesis and mitochondrial function in NASH, and prevents NASH progression in vivo. Thus, the disrupted DHPS-DOHH-EIF5A[H] pathway during NASH represents a therapeutic target to increase hepatic protein synthesis and mitochondrial fatty acid oxidation (FAO) and prevent NASH progression.

Polyamine biosynthesis is a conserved metabolic pathway present in all organisms[1]. Besides the endogenous biosynthesis pathway, polyamines are obtained from dietary sources and microbial production in the gut[2,3]. Polyamines, particularly spermidine, extend the lifespan of several species, including yeast, flies, worms, and mice[4,5]. Spermidine also exerts cardio- and neuroprotective effects during aging[4–7]. In mice, supplementation of spermidine reduces aging-associated cardiac hypertrophy, preserves diastolic function, and improves cognitive function[5–7]. In humans, dietary spermidine intake reduced the risk for cardiovascular diseases and cognitive impairment[5,7]. The anti-aging effects of spermidine have been mainly

attributed to its induction of autophagy[4,5,7], a protective cellular process that declines with aging[8].

Spermidine also serves as the sole substrate for hypusination of a lysine residue in eukaryotic initiation factor 5 A (EIF5A), which is critical for protein synthesis. EIF5A is the only known protein requiring this rare, but essential, post-transcriptional modification. Hypusination is a two-step enzymatic reaction involving deoxyhypusine synthase (DHPS) and deoxyhypusine hydroxylase (DOHH) that occurs during or shortly after the synthesis of EIF5A (Fig. 1a). Hypusinated EIF5A (EIF5A[H]) regulates protein synthesis through its involvement in translation initiation, elongation, and termination[9,10]. The majority of

**Fig. 1 | Decreased DHPS-DOHH-EIF5A^H pathway in NAFLD. a** Polyamine synthesis and hypusination of EIF5A pathway in eukaryotic cells. Enzymes *ARG1*, *ODC*, *SRM* are involved in converting arginine to spermidine. *DHPS* and *DOHH* use spermidine to hypusinate EIF5A. **b** Violin plots showing mRNA levels of genes involved in endogenous polyamine biosynthesis and EIF5A hypusination in Control (*n* = 19), steatosis (*n* = 10), and NASH (*n* = 16) from publicly available database (accession number E-MEXP-3291, http://www.webcitation.org/5zyojNu7T)[20]. **c** Violin plots showing mRNA levels of genes involved in endogenous polyamine biosynthesis and EIF5A hypusination in Control (*n* = 12), steatosis (*n* = 9), and NASH (*n* = 17) from publicly available database (GSE48452)[21]. **b**, **c** Significance was calculated by one-way ANOVA or Kruskal–Wallis test, as appropriate. **d** Quantitative-PCR analysis of

mRNA levels of polyamine metabolism genes in the livers from mice fed with NCD (*n* = 8) or WDF (*n* = 8) for 16 weeks. **e** Western blot and densitometric analysis of protein levels of Dhps, Dohh, eIF5A^H, and eIF5A in the liver from mice fed with NCD (*n* = 7) or WDF (*n* = 6) for 16 weeks. **f**, **g** mRNA expression of genes in polyamine biosynthesis and hypusination pathways (**f**, *n* = 5), and protein levels of Dhps, Dohh, eIF5A^H, and eIF5A (**g**, *n* = 3) in AML12 hepatic cells treated with fatty acids (FA, palmitic acid 0.6 mM, oleic acid 0.17 mM) for 48 h. **f**–**g** Data were shown as box-and-whisker with median (middle line), 25th–75th percentiles (box), and min-max values (whiskers). **d**–**g** significance was calculated by two-tailed Student's *t* test or Mann–Whitney *U* test, as appropriate. Source data are provided as a Source Data file.

cellular EIF5A is hypusinated during normal physiological conditions[11–13]. Although hypusination of EIF5A is conserved throughout evolution[14], currently little is known about its dysfunction in human diseases.

Non-alcoholic fatty liver disease (NAFLD) is the most common chronic liver disease with global prevalence estimated to be ~25% of all adults[15]. NAFLD covers a spectrum of liver diseases, ranging from simple steatosis to non-alcoholic steatohepatitis (NASH), characterized by progressive inflammation, hepatocyte injury, and fibrosis[16].

Multiple parallel hits have been postulated for the pathogenesis of NAFLD[17]; however, the complex molecular mechanisms underlying disease progression are not well understood[18,19]. Currently, there are no FDA-approved drugs for NAFLD/NASH. The management for NAFLD mainly includes diet and lifestyle changes for achieving weight loss and management of underlying metabolic risk factors.

Here, we report decreased hepatic EIF5A^H due to impairment of the DHPS-DOHH-EIF5A^H pathway in patients and mice with NAFLD, and hepatic cells subjected to lipid accumulation. The decrease in EIF5A^H

led to decreased synthesis of mitochondrial proteins, reduced mitochondrial activity, and diminished β-oxidation of fatty acids (FA). Spermidine treatment preserved Eif5a[H] in a mouse model of NASH, increased β-oxidation of FA, and partially prevented hepatosteatosis, inflammation, and fibrosis associated with NASH.

## Results

### Decreased DHPS-DOHH-EIF5A[H] pathway in NAFLD

We first examined the hepatic mRNA levels of enzymes involved in polyamine biosynthesis/metabolism pathways (Fig. 1a) in patients with NAFLD from two published datasets[20,21]. Both datasets contained normal subjects, and patients with steatosis or NASH. There were no consistent changes observed in the two datasets for spermidine biosynthesis enzymes such as arginase (*ARG1*), ornithine decarboxylase (*ODC1*), spermidine synthase (*SRM*), and spermidine/spermine N (1) acetyltransferase (*SAT1*) (Fig. 1a–c). For enzymes that utilize spermidine as substrate to promote EIF5A hypusination (EIF5A[H]), *DHPS* was decreased in patients with NASH in one dataset, and *DOHH* was decreased in patients with NASH in both datasets (Fig. 1b, c). These results suggested there was impairment of the DHPS-DOHH-EIF5A[H] pathway in patients with NASH. We also examined these pathways in patients with mild (fibrosis stage 0–1) and severe NAFLD (fibrosis stage 3–4), and found that patients with severe NAFLD showed decreased *ARG1* and increased *SMS* mRNA levels, which may cause decreased spermidine synthesis and increased conversion of spermidine to spermine (Fig. S1). *DHPS* also was decreased in patients with severe NAFLD (Fig. S1). All these observed changes suggested there could be decreased EIF5A[H] in patients with severe NAFLD.

We next investigated the Dhps-Dohh-Eif5a[H] pathway in a dietary model of NASH in which mice were fed a western diet supplemented with liquid fructose (WDF)[22]. After 16 weeks, these mice developed obesity, insulin resistance, and NAFLD that resembled the human phenotype of early NASH[22]. The hepatic expression of polyamine biosynthesis and hypusination pathway genes in mice fed with WDF showed decreased hepatic *Arg1*, *Srm*, *Dhps*, and *Dohh* at 16 weeks (Fig. 1d), and Dhps, Dohh, and Eif5a[H] protein were decreased at 16 weeks compared to control mice fed normal chow diet (NCD) (Fig. 1e, Fig. S2a). Taken together, these results demonstrated there was a decrease in the polyamine synthesis and Dhps-Dohh-Eif5a[H] pathway in this mouse model of NASH.

We next treated cultured AML12 hepatic cells with palmitic and oleic FA for 48 h, and found that FA decreased *Odc1*, *Srm*, *Dhps*, and *Dohh* mRNA expression (Fig. 1f). FA also decreased Dhps, Dohh, Eif5a, and Eif5a[H] protein levels relative to control cells (Fig. 1h, Fig. S2a). These results suggested that the accumulation of intracellular lipid impaired both the polyamine biosynthesis and the Dhps-Dohh-Eif5a[H] pathways in hepatic cells.

### Eif5a[H] regulates protein synthesis and mitochondrial function in hepatic cells

To understand the role of endogenous Eif5a[H] in hepatic cells, we employed *Dohh* or *Eif5a* siRNA to knockdown their expression in AML12 cells and found that *Dohh* or *Eif5a* knockdown (KD) decreased Eif5a[H] levels (Fig. 2a). We then used an acute puromycin incorporation assay to measure protein synthesis rate[23], and found that *Dohh* or *Eif5a* KD decreased general protein synthesis rate (Fig. 2a, Fig. S3). Previously, it was suggested that Eif5a[H] facilitated the synthesis of several mitochondrial proteins[24]. Accordingly, we examined the protein expression of key transcription factors that regulated mitochondrial biogenesis (Tfam and PGC1α (encoded by *Ppargc1a* gene)), and mitochondrial proteins including outer membrane proteins (Vdac1, Tomm20), subunits of electron transfer chain enzymes (Ndufb8, Sdhb, cytochrome C (Cyt C, encoded by *Cycs* gene), CoxIV (encoded by *Cox4i1* gene) and ATP synthase (Atp5a, encoded by *Atp5a1* gene). Interestingly, Tfam, PGC1α, Vdac1, Tomm20, Sdhb, Cyto C, and CoxIV

protein levels were decreased in both *Dohh* or *Eif5a* KD cells (Fig. 2b) whereas Ndufb8 and Atp5a protein expression decreased only in *Eif5a* KD cells. We examined the effect of *Dohh* or *Eif5a* KD on the gene expression of these mitochondria-related proteins, and found that with the exception of *Vdac1*, none of the other genes had decreased transcription with siRNA KD (Fig. S4). These findings strongly suggested the decreases in mitochondria-related protein expression in *Dohh* or *Eif5a* KD cells were likely due to reduced Eif5a[H] and its subsequent effect(s) on protein synthesis.

We then performed LC-MS/MS proteomic analysis on pooled samples to identify hepatic proteins that were differentially expressed due to *Dohh* KD in AML12 cells. Among the 4286 proteins quantified in both control and *Dohh* KD cells, 297 proteins were up-regulated and 290 proteins were down-regulated (>1.5-fold changes and $p < 0.05$) in *Dohh* KD cells (Supplementary Table 1). Due to the lack of replicates from proteomics data, we were not able to compare the quantification of individual proteins. Instead, we performed GO ontology pathway analysis (Fig. 2c, Fig. S5, Supplementary Data 1 and 2). Down-regulated proteins were enriched for cellular components including ribosomal and mitochondrial proteins, and biological processes showed that down-regulated proteins were enriched for ribosome biogenesis, and rRNA metabolic process (Fig. 2c). Thus, pathway analyses of the proteome suggested that proteins involved in ribosomal or mitochondrial protein synthesis were down-regulated by *Dohh* KD.

We next examined the effects of *Dohh* or *Eif5a* KD on mitochondrial function by Seahorse XF Cell Mito Stress test. Both *Dohh* or *Eif5a* KD decreased mitochondrial basal respiration and ATP turnover, and only *Eif5a* KD decreased maximal respiration (Fig. 2d). Consistent with these findings of decreased mitochondrial activity, we found there was decreased mitochondria DNA copy number (mtDNA) in *Dohh* or *Eif5a* KD cells compared to control cells (Fig. 2e). Taken together, these results showed that loss of Dohh or Eif5a in hepatic cells led to decreased Eif5a[H] levels and protein synthesis, reduced mitochondrial proteins and DNA, and diminished mitochondrial oxidative phosphorylation (OXPHOS).

### Spermidine supplementation preserves Eif5a[H] and protein synthesis in FA-treated cells

Since FA-treated hepatic cells displayed impaired Dhps-Dohh-Eif5a[H] pathway (Fig. 1f–h), we examined whether FA impaired protein synthesis in hepatic cells. Indeed, we found a decrease in acute puromycin incorporation of low to high molecular weight proteins in AML12 cells treated with FA, suggesting there was impairment of general protein synthesis rate in FA-treated cells (Fig. 3a, Fig. S6). spermidine co-treatment with FA (FA + Spd) restored Eif5a[H], and partially restored puromycin incorporation (Fig. 3a), indicating that spermidine co-treatment partially restored general protein synthesis rate in FA-treated cells. Spermidine also restored protein level of Eif5a without increasing its mRNA level (Fig. 3a and Fig. S7a), indicating that spermidine may increase the synthesis of Eif5a protein in FA-treated cells. We next measured protein levels of Tfam, PGC1α, Vdac1, Tomm20, Ndufb8, Sdhb, Cyto C, and CoxIV mitochondrial proteins, and found they were decreased in FA-treated cells (Fig. 3b). The mRNA levels of *Tfam*, *Vdac1*, *Tomm20*, and *Sdhb* were decreased whereas *Ppargc1a*, *Ndufb8*, *Cycs*, and *Cox4i1* were unchanged (Fig. S7b). Thus, the expression of one subgroup of mitochondrial proteins was affected by decreased transcription, whereas another group was primarily by decreased protein synthesis with the presence of FA. Interestingly, spermidine co-treatment with FA restored most of the examined mitochondrial proteins without affecting their mRNA levels (Fig. 3b and Fig. S7c), suggesting that spermidine-mediated Eif5a[H] primarily increased the synthesis of mitochondrial proteins in FA-treated cells.

We next performed proteomic and pathway analysis in AML12 cells treated with media containing either no FA (control), FA, or FA with spermidine co-treatment (FA + Spd). FA-treated cells showed 254

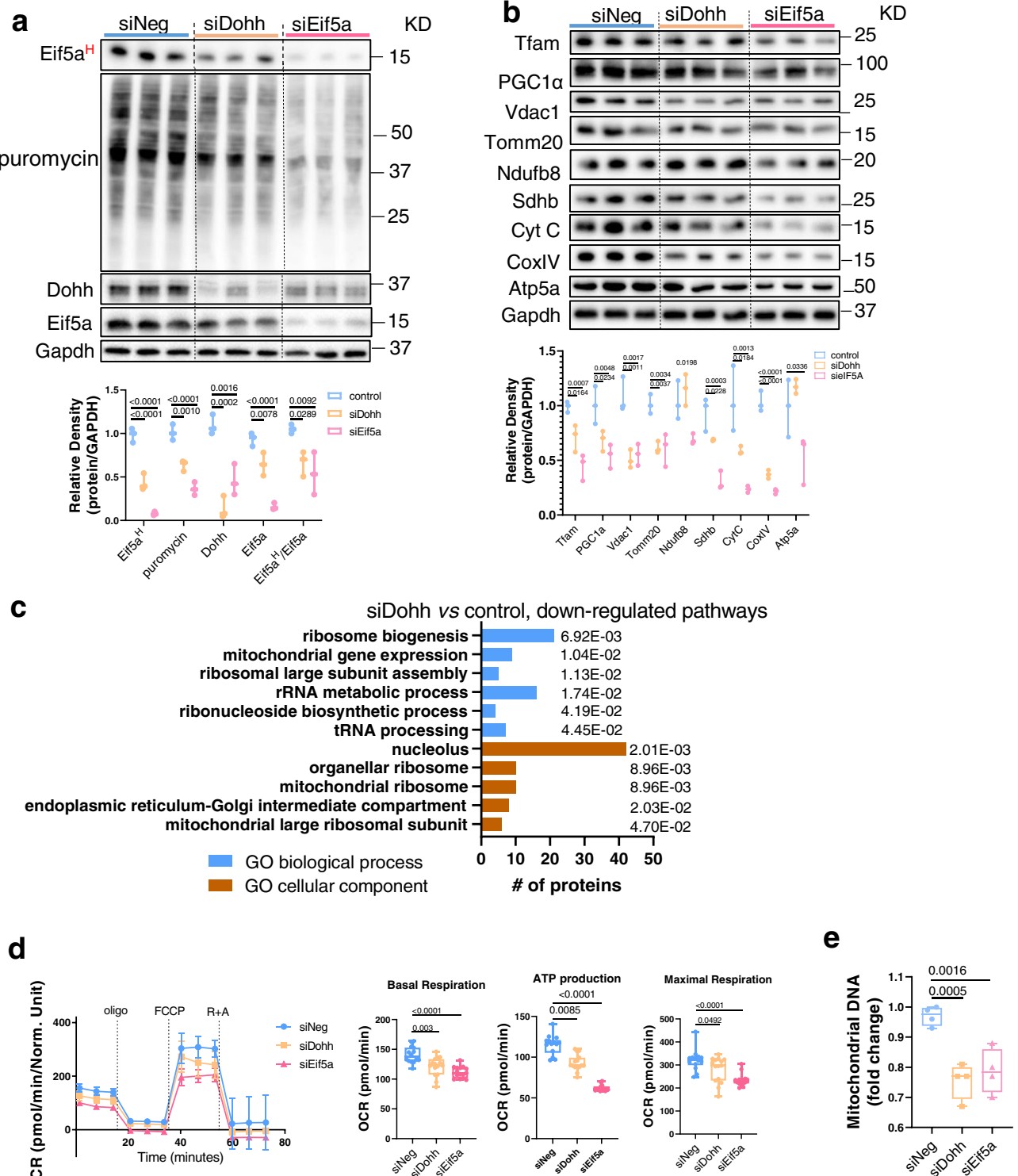

**Fig. 2 | Dohh KD decreased protein synthesis and mitochondrial function in hepatic cells.** AML12 cells were transfected with 20 nM of negative (siNeg), Dohh (siDohh), or eIF5A (siEif5a) siRNA for 48 h. Puromycin (1 μg/ml) was added 1 h before harvesting cell lysate. **a** Western blot and densitometric analysis of eIF5A[H], and proteins after puromycin incorporation. (*n* = 3). **b** Western blot and densitometric analysis of Tfam, PGC1α, and mitochondrial proteins. (*n* = 3). **c** Proteomics and gene ontology (GO) enrichment analysis (with corrected *p* value indicated on the bar) of downregulated proteins in Dohh KD (siDohh) vs control AML12 cells. A

Bonferroni correction was applied to correct for multiple testing. **d** Agilent Seahorse XF Mito Stress Test measurement of mitochondrial OXPHOS. Oligomycin (oligo), FCCP, rotenone (R), and antimycin A (A) were used at 1 μM each. (*n* = 15). Line graph is presented as mean value ± SEM. **e** Mitochondrial DNA copy number in Dohh or eIF5A knockdown cells. (*n* = 4). **a**, **b**, **d**, **e** Data were shown as box-and-whisker with median (middle line), 25th–75th percentiles (box), and min-max values (whiskers). Significance was calculated by one-way ANOVA or Kruskal–Wallis test, as appropriate. Source data are provided as a Source Data file.

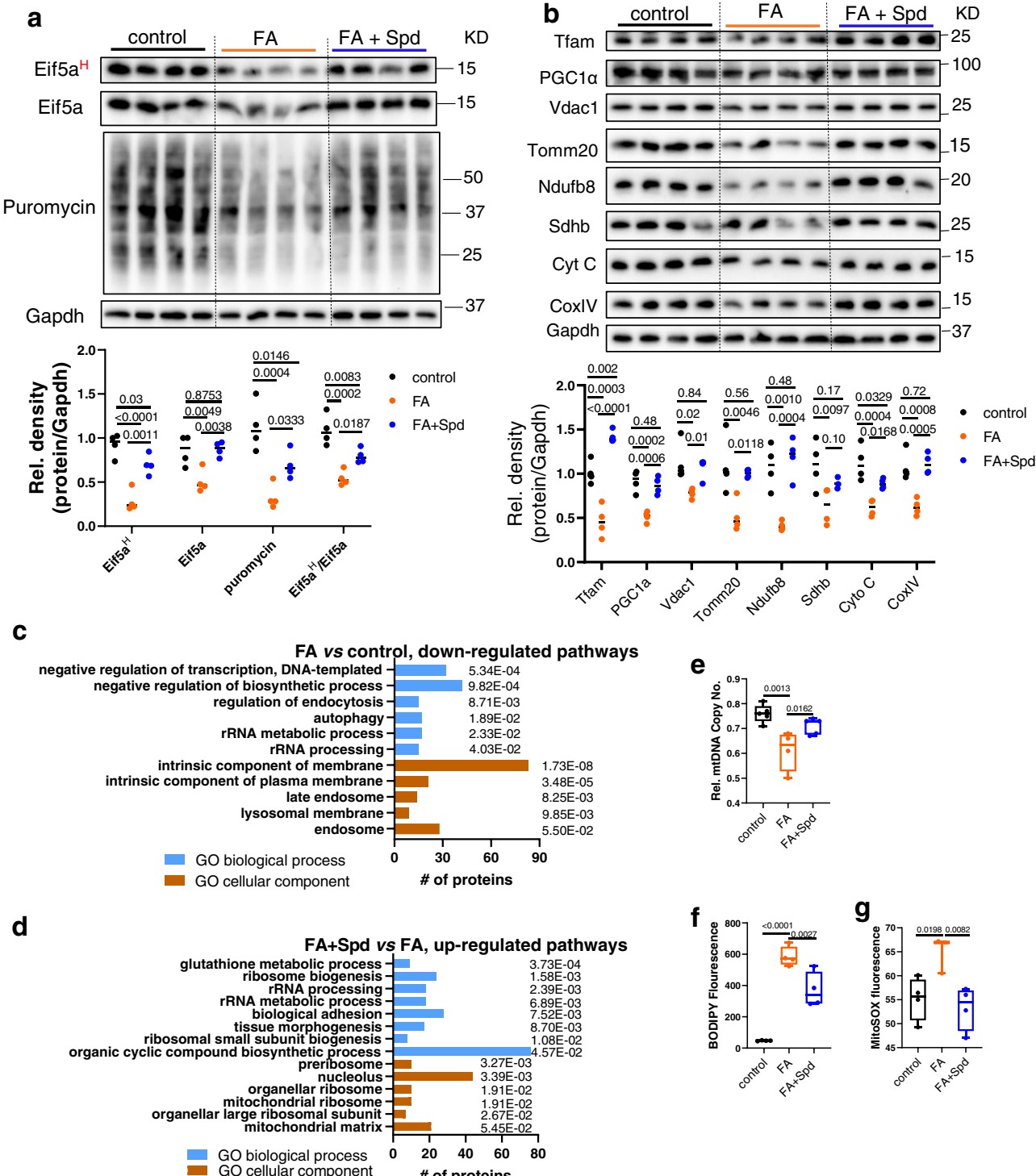

**Fig. 3 | FA decreased Eif5a$^H$, protein synthesis rate and mitochondrial proteins, which can be rescued by co-treatment with spermidine. a, b** Western blot and densitometric analysis of Eif5a$^H$ and total proteins after puromycin incorporation (**a**), and mitochondrial proteins (**b**) in control, FA−, and FA+ Spd-treated AML12 cells. (*n* = 4). **c, d** Proteomics and gene ontology (GO) enrichment analysis (with corrected *p* value indicated on the bar) of downregulated proteins in FA-treated *vs* control AML12 cells (**c**), and upregulated proteins in FA+ Spd-treated *vs* FA-treated AML12 cells (**d**). A Bonferroni correction was applied to correct for multiple testing. **e**–**g** Relative mitochondrial copy number ((**e**, control (*n* = 5), FA (*n* = 4), FA + Spd

(*n* = 5)), BODIPY fluorescence (**f**, *n* = 4), and MitoSOX measurement of mitochondrial ROS ((**g**, control (*n* = 4), FA (*n* = 3), FA + Spd (*n* = 4)) in control, FA−, FA + Spd-treated AML12 cells. AML12 cells were treated with BSA-conjugated FA (palmitic acid 0.6 mM, oleic acid 0.17 mM) with or without spermidine (Spd, 100 μM) for 48 h. **e**–**g** Data were shown as box-and-whisker with median (middle line), 25th–75th percentiles (box), and min-max values (whiskers). Significance was calculated by one-way ANOVA or Kruskal–Wallis test as appropriate. Source data are provided as a Source Data file.

upregulated and 332 down-regulated proteins when compared to control cells (>1.5-fold changes and $p < 0.05$). Pathway analysis (Fig. 3c, Fig. S8, Supplementary Data 3−6) of cellular components showed that down-regulated proteins were enriched for endosome, and membrane proteins (Fig. 3c), whereas pathway analysis of biological processes revealed rRNA processing were impaired in FA-treated cells. In contrast, cellular component analysis of up-regulated proteins from FA + Spd-treated cells showed enrichment of mitochondrial and ribosomal proteins, whereas pathways of biological process revealed ribosome biogenesis and rRNA processing were increased compared to FA-treated cells (Fig. 3d). These latter findings were consistent with the up-regulated protein synthesis rates that we observed previously in FA + Spd-treated cells (Fig. 3a).

We next assessed the effects of FA and spermidine on mtDNA, intracellular lipid content, and mitochondrial reactive oxygen species (ROS), and found that FA reduced mtDNA, and increased BODIPY and MitoSOX fluorescence whereas spermidine partially reversed both mtDNA and BODIPY fluorescence (Fig. 3e, f), and fully reversed MitoSOX fluorescence (Fig. 3g). These data showed that spermidine also restored mitochondrial number, decreased intracellular lipid content and ROS in FA-treated cells.

### Spermidine restores protein synthesis and mitochondrial function in an Eif5a^H-dependent manner

It has been demonstrated previously that spermidine can be oxidized by FBS-containing amine oxidases and cause cytotoxicity in cultured cancer cells[25]. To rule out the possibility that the observed effect of spermidine on mitochondrial proteins were due to FBS-containing amine oxidases oxidized spermidine products, we cultured AML12 cells using a serum-free medium, and treated cells with no FA (control), FA and FA + Spd. We have found that in serum-free medium, spermidine supplementation was able to restore levels of Eif5a^H and mitochondrial proteins (Fig. S9). Next, we determined whether spermidine's beneficial effects in FA-treated cells on protein synthesis and mitochondrial proteins were mediated by Eif5a^H. Accordingly, we employed Dohh siRNA KD to block spermidine's rescue of Eif5a^H, and found that spermidine's stimulation of general protein synthesis rate and mitochondrial protein expression were dependent upon Eif5a^H (Fig. 4a, b, Fig. S10). Proteomics and pathway analysis of cellular components and biological processes (Fig. 4c, d, Fig. S11, Supplementary Data 7, 8) showed that Dohh KD abrogated spermidine's rescue of nucleolus protein and ribosomal assembly in FA-treated cells (Fig. 4c, d). Approximately 22.2 to 24.1% of these proteins were co-regulated as they were up-regulated by spermidine and decreased by Dohh KD. Spermidine also increased mitochondrial OXPHOS in FA-treated cells; however, it failed to do so in Dohh KD cells under the same conditions (Fig. 4e). These results demonstrated that spermidine increased mitochondrial OXPHOS in an Eif5a^H-dependent manner in FA-treated cells.

Autophagic removal of damaged mitochondria (mitophagy) in conjunction with mitochondrial biosynthesis is another mechanism for maintaining mitochondrial quality and activity[26,27]. Spermidine induced autophagy in several tissues[4,5] and likely involved Eif5a^H-mediated synthesis of several autophagy-related proteins, including the key transcription factor regulates autophagy, transcription factor EB (TFEB)[28,29]. Accordingly, we assessed the effects of FA treatment on autophagy and found that it reduced TFEB and LC3B-II proteins levels; however, these effects could be prevented by spermidine. Of note, spermidine failed to restore the levels of TEFB and LC3B-II in Dohh KD cells (Fig. S12). Thus, our results suggested that Eif5a^H-dependent restoration of autophagy in FA-treated cells may be an additional mechanism for spermidine's beneficial effect on mitochondrial activity. In addition, spermidine's restoration of Eif5a protein expression was blunted by Dohh KD (Fig. 4a), suggesting that the synthesis of Eif5a also depended upon Eif5a^H.

### Spermidine prevents the loss of hepatic Eif5a^H, mitochondrial protein, and NAFLD progression

Given spermidine's capability to restore protein synthesis rate and mitochondrial number/activity in FA-treated hepatic cells, we investigated spermidine's action(s) on the progression of NAFLD in vivo in mice fed WDF without or with 3 mM spermidine supplementation (WDF + Spd) in their drinking water for 16 weeks (Fig. 5a). H&E staining showed that spermidine reduced the hepatic lipid droplet size and number in mice fed WDF for 16 weeks (Fig. 5b). Measurement of total hepatic triglyceride (TAG) content confirmed that WDF increased hepatic TAG content and spermidine supplementation of WDF partially reversed this increase (Fig. 5c). Lipidomic profiling employing liquid chromatography coupled with mass spectrometry (LC-MS) demonstrated that WDF increased hepatic TAGs, diacylglycerols (DAGs), monoacylglycerols (MAGs), and cholesterol ester (CEs) levels. Spermidine supplementation partially reversed these changes in hepatic lipid species (Fig. 5d−g, Supplementary Data 9).

Spermidine decreased serum alanine aminotransferase (ALT) level (Fig. 5h) and hepatic mRNA levels of inflammatory markers such as Il1b, Ccl2, and Cxcl9 (Fig. S13), suggesting that spermidine reduced hepatic inflammation and damage. Since fibrosis is a major feature of NASH progression[30], we assessed hepatic collagen content by measuring the intrahepatic level of hydroxyproline, a signature amino acid that comprises approximately 13.5% of the amino acids in fibrillar collagens[31]. Hepatic hydroxyproline content and Masson trichrome staining for collagen were increased in mice fed WDF and partially reversed when mice fed WDF were supplemented with spermidine (Fig. 5i, j). Taken together, our data demonstrated that spermidine supplementation of WDF prevented hepatosteatosis and hepatic inflammation, damage, and fibrosis.

In mice fed WDF, Dhps, Dohh, and Eif5a^H protein expression decreased compared to mice fed NCD. Eif5a protein level was unchanged (Fig. 6a, c, Fig. S14). Spermidine did not change Dhps and Eif5a protein levels in mice fed WDF but preserved Dohh protein expression and Eif5a^H/Eif5a at levels similar to those in mice fed NCD (Fig. 6b, c). Interestingly, spermidine also increased hepatic Dohh mRNA expression in mice fed WDF without changing the mRNA levels of the enzymes involved in polyamine synthesis (Fig. S15a). In addition, the expression levels of proteins involved in mitochondrial biogenesis and function: Tfam, Tomm20, Ndufb8, Sdhb, Cyto C, CoxIV, Atp5a, and Uqcr2, were decreased in mice fed WDF and were unchanged or only partially decreased in mice fed WDF supplemented with spermidine when compared to mice fed NCD (Fig. 5d−f). The decreases in mitochondrial protein expression in mice fed WDF often were associated with decreases in their mRNA expression suggesting there was a contribution by transcriptional regulation (Fig. S15b). However, spermidine did not change Vdac1, Tomm20, Sdhb, Cycs, Cox4i1, and Uqcrc2 mRNA levels (Fig. S15c), supporting the notion that spermidine up-regulated their protein expression post-transcriptionally. On the other hand, spermidine increased Tfam, Ndufb8 and Atp5a1 mRNA levels (Fig. S15c), suggesting that spermidine increased the gene expression of some mitochondrial proteins in vivo. There were no significant differences in mitochondrial DNA content and serum β-hydroxybutyrate levels between control fed NCD and mice fed WDF; however, mice fed WDF supplemented with spermidine increased mitochondrial DNA content and serum β-hydroxybutyrate levels compared to mice fed NCD or WDF (Fig. 6g, h). Taken together, our results demonstrated that spermidine supplementation of mice fed WDF prevented or partially prevented reductions in hepatic Eif5a^H and mitochondrial protein levels, increased mitochondrial biogenesis and β-oxidation of FA, reduced hepatic acylglycerol and cholesterol levels, and partially prevented hepatosteatosis and NASH progression in vivo in a dietary model of NASH.

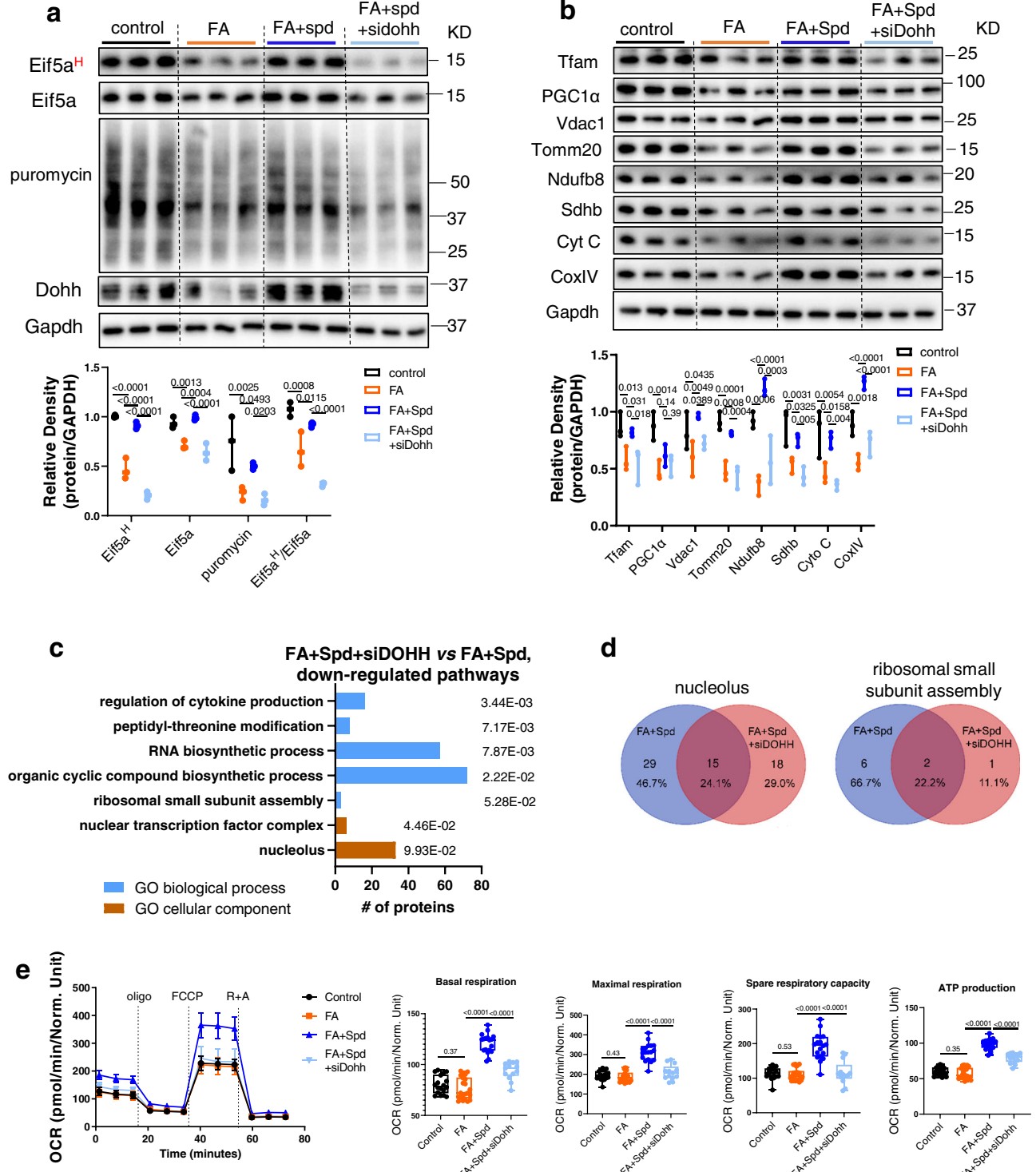

**Fig. 4 | Spermidine rescued mitochondrial function in Eif5a[H]-dependent manner. a, b** Western blot and densitometric analysis of Eif5a[H], and puromycin corporation (**a**, *n* = 3), and mitochondrial proteins (**b**, *n* = 3). **c** Proteomics and gene ontology (GO) enrichment analysis (with corrected *p* value indicated on the bar) of downregulated proteins in FA + Spd+siDohh *vs* FA + Spd cells. A Bonferroni correction was applied to correct for multiple testing. **d** Venn diagram showing the overlap of proteins under identified pathways in 4c and 3d. **e** Agilent Seahorse XF Mito Stress Test of mitochondrial OXPHOS. AML12 cells were first transfected with

20 nM of negative (siNeg), or Dohh (siDohh) siRNA for 24 h, followed by treatment with BSA-conjugated FA (palmitic acid 0.6 mM, oleic acid 0.17 mM) with or without spermidine (Spd, 100 μM) for 48 h. Line graph is presented as mean value ± SEM. (control (*n* = 18), FA (*n* = 22), FA + Spd (*n* = 18), FA + Spd+siDohh (*n* = 13)) **a, b, e** Data were shown as box-and-whisker with median (middle line), 25th–75th percentiles (box), and min-max values (whiskers). Significance was calculated by one-way ANOVA or Kruskal–Wallis test, as appropriate. Source data are provided as a Source Data file.

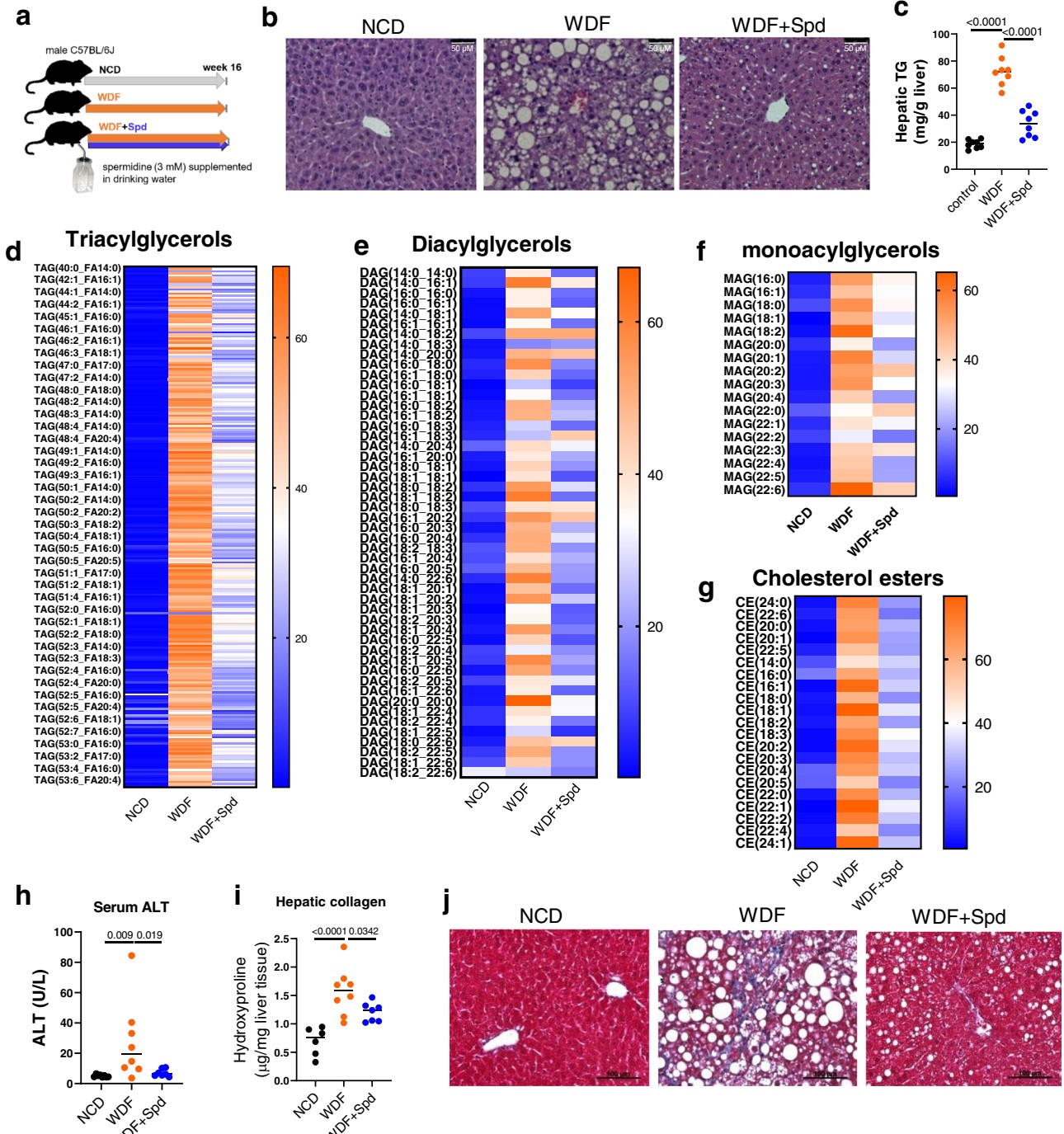

**Fig. 5 | Spermidine supplementation prevented NAFLD progression in a dietary mouse model of NASH. a** Schematic showing administration protocol for spermidine (3 mM in drinking water) in mice fed with WDF for 16 weeks. **b** Representative images of H&E staining of the liver. (*n* = 5). **c** Hepatic triglyceride content was measured by biochemical assay. (*n* = 8). **d–g** Liquid chromatography coupled with mass spectrometry (LC-MS) lipidomic profiling of hepatic triacylglycerols (TAGs) (**d**), diacylclycerols (DAGs) (**e**), monoacylglycerol (MAGs) (**f**), and cholesterol esters (CEs) (**g**) levels from mice fed with NCD (*n* = 6), WDF (*n* = 8), or WDF supplemented with spermidine (WDF + Spd, *n* = 6) for 16 weeks. Heat maps represent the average of each condition. **h** circulating ALT level from mice fed with NCD (*n* = 8), WDF (*n* = 8), WDF + Spd (*n* = 7). **i** Hepatic collagen content was determined by measurement of hydroxyproline level in the liver from mice fed with NCD (*n* = 6), WDF (*n* = 8), WDF + Spd (*n* = 7). **j** Representative images of Masson trichrome stain of liver sections. (*n* = 5). Significance was calculated by One-Way ANOVA. Source data are provided as a Source Data file.

## Discussion

The overlapping polyamine biosynthesis and DHPS-DOHH-EIF5A[H] pathways (Fig. 1a) are highly conserved in eukaryotic cells[1]. Dhps-null mice are embryonic lethal[13], and emphasize the essential role of these pathways for prenatal survival. Despite their importance, little is known about the roles of polyamine-EIF5A[H] pathways on liver function in the normal state or disease conditions. Here, we report that the DHPS-DOHH-EIF5A[H] pathway is dysregulated in NAFLD. Patients with NASH had decreased hepatic mRNA expression of the key enzyme *DOHH* in this pathway (Fig. 1b, c). Similarly, using a dietary model of NAFLD in which mice were fed WDF 16 weeks, we observed impairment of the polyamine biosynthesis and Dhps-Dohh-Eif5a[H] pathways

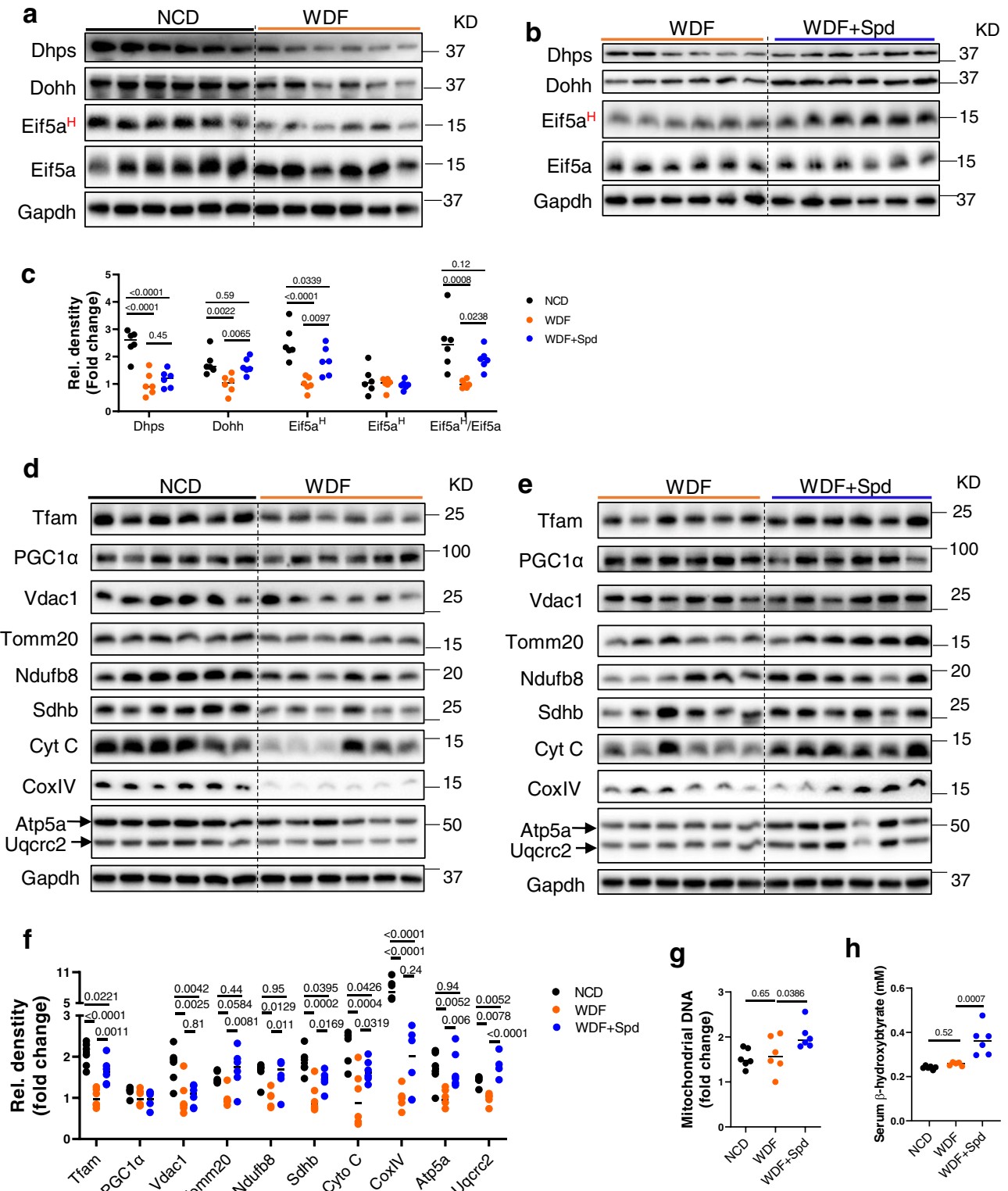

**Fig. 6 | Spermidine supplementation restored hepatic Eif5a[H] and mitochondrial protein levels in a dietary mouse model of NASH. a–c** Western blot and densitometric analysis of hepatic Dhps, Dohh, Eif5a[H], and Eif5a in NCD vs. WDF (**a**), and WDF vs. WDF + Spd (**b**) mice. The blots in a and b were processed in parallel. Densitometric analysis (**c**) was first normalized with GAPDH, and then calculated the fold change against WDF (NCD *vs* WDF, and WDF + Spd vs WDF). (*n* = 6) (**d–f**) Western blot and densitometric analysis of Tfam, PGC1α, and mitochondrial

proteins in the liver from mice fed with NCD vs. WDF (**d**), or WDF vs. WDF + Spd (**e**). The blots in **d** and **e** were processed in parallel. Densitometric analysis (**f**, *n* = 6) was first normalized with GAPDH, and then calculated the fold change against WDF (NCD vs WDF, and WDF + Spd *vs* WDF). (**g, h**) Mitochondrial DNA copy number (**g**) and circulating β-hydroxybutyrate (**h**) in NCD (*n* = 6), WDF (*n* = 6), and WDF + Spd (*n* = 6) groups. Significance was calculated by one-way ANOVA or Kruskal–Wallis test, as appropriate. Source data are provided as a Source Data file.

exemplified by decreased levels of Dhps and Dohh mRNA and protein, and Eif5a[H] protein during early NASH (Fig. 1d, e).

We then performed *Dohh* and *Eif5a* siRNA KD in mouse hepatic AML12 cells, and found they decreased Eif5a[H] protein expression and general protein synthesis rate, and reduced levels of mitochondrial protein expression (Fig. 2a, b). Proteomic/pathway analysis of *Dohh* KD cells showed that pathways involved in mitochondrial biosynthesis, and ribosome-mediated protein synthesis were down-regulated (Fig. 2c). *Dohh* and *Eif5a* KD also impacted mitochondrial function as they both decreased mitochondrial OXPHOS (Fig. 2d). These findings demonstrated the role of endogenous Eif5a[H] in synthesis of mitochondrial proteins and maintaining mitochondrial function. Since FA also decreased mitochondrial function and increased ROS generation in hepatic cells[32], we next investigated the involvement of Eif5a[H] in FA-induced mitochondrial dysfunction, and found that FA caused a down-regulation of Dhps-Dohh-Eif5a[H] pathway, general protein synthesis rate, and mitochondrial protein expression (Fig. 3). Spermidine supplementation reversed or partially reversed FA's effect on Eif5a[H], general protein synthesis, and mitochondrial protein expression (Fig. 3). Proteomic/pathway analysis showed that spermidine co-treatment with FA up-regulated cellular processes involving ribosome biogenesis, and protein synthesis (Fig. 3d). Spermidine supplementation also increased mitochondria DNA copy number, decreased intracellular triglyceride content, and decreased mitochondrial ROS (Fig. 3e–g), and concomitantly improved mitochondrial OXPHOS (Fig. 4e). *Dohh* KD abrogated the beneficial effects of spermidine on Eif5a[H], protein synthesis, and mitochondrial OXPHOS (Fig. 4). Taken together, our data showed an essential role of Dhps-Dohh-Eif5a[H] pathway in the regulation of mitochondrial protein and metabolic function. We also showed that impairment of Dhps-Dohh-Eif5a[H] pathway and its associated reduction in protein synthesis contributed to mitochondrial dysfunction in FA-treated hepatic cells.

Defects in mitochondrial morphology, lower mitochondrial respiratory chain activity, and decreased ATP synthesis previously have been described in NAFLD[33], and likely contribute to altered lipid oxidation and liopotoxicity in NAFLD[33,34]. Furthermore, dysfunctional mitochondria may contribute to the progression of hepatic inflammation, injury, and fibrosis of NAFLD/NASH[35–37]. We thus investigated the role of Dhps-Dohh-Eif5a[H] pathway on mitochondrial protein expression in vivo, and found a decreased abundance of OXPHOS proteins in mice with NASH as previously reported[33,38]. Importantly, decreased mitochondrial protein expression was associated with impaired hepatic Dhps-Dohh-Eif5a[H] pathway in NASH. Spermidine supplementation of WDF prevented the decreases in Dohh and Eif5a[H] protein levels, and increased the synthesis of several mitochondrial proteins and mitochondrial number (Fig. 6, Fig. S15). Of note, spermidine supplementation also increased fatty acid oxidation (FAO) (Fig. 6h), partially reduced hepatic lipid species, and partially prevented hepatic steatosis, inflammation, and fibrosis in mice fed WDF (Fig. 5). Taken together, our results showed that spermidine supplementation prevented or partially prevented the attenuation of Eif5a[H] and mitochondrial protein expression in mice fed WDF, and increased mitochondrial FAO to reduce lipotoxicity and NASH progression. Emerging evidence has demonstrated the beneficial effect of spermidine in preventing hepatosteatosis and fibrosis[39–41]. Ma et al. reported that 8-week spermidine supplementation in preexisting obese mice decreased hepatic steatosis and circulating ALT levels. Spermidine decreased hepatic expression of genes involved lipogenesis, and increased gene expression of PPARα and CPT1α, key players promoting lipid oxidation, as well as antioxidant enzymes[39]. Spermidine supplementation also reduced fibrosis through MAP1S-mediated autophagy and NRF2 signaling in carbon tetrachloride (CCl4)-induced liver fibrosis[40,41]. Taken together, we believe that spermidine may exert beneficial effects against hepatic steatosis and fibrosis in NAFLD and other diseases, and future investigation in man is warranted.

Recently, EIF5A[H]-dependent synthesis of mitochondrial proteins has been described in the brain and immune cells[6,7,24], although the mechanism(s) by which it regulates the synthesis of mitochondrial proteins is not fully understood. It has been reported that EIF5A[H] is required for the elongation of consecutive polyproline motifs, conserved non-polyproline tripeptide sequences[10,42–46], and mitochondrial targeting sequences harbored by mitochondrial proteins[24]. Similarly, we observed that the expression of several mitochondrial proteins decreased in parallel with reduced EIF5A[H] without any changes in their mRNA levels in FA-treated hepatic cells and livers from mice with NASH (Figs. 2–4, 6, Figs. S4, S7, S15). Although we cannot exclude other post-transcriptional mechanism(s), such as protein stability, our data showing dependence of their protein synthesis upon EIF5A[H] suggests that the synthesis of some mitochondrial proteins (e.g., Tomm20, Ndufb8, Sdhb, CoxIV), in addition to transcription of others, play an important role in maintaining mitochondrial number and function. Additionally, EIF5A[H] may regulate mitochondrial biogenesis indirectly via upregulation of Tfam. In support of this notion, Dohh KD and the resultant decrease in EIF5A[H] led to reduced expression of Tfam (Fig. 2), whereas spermidine supplementation increased EIF5A[H] in hepatic cells treated with FA and in livers from mice fed WDF, and led to increased Tfam (Figs. 3, 4, 6). Finally, spermidine reportedly induced autophagy in several tissues[4,5]. One mechanism may be through eIF5A[H]-mediated synthesis of autophagy-related proteins such as the key transcription factor TFEB[28,29]. We also showed that spermidine increased autophagy in FA-treated hepatic cells (Fig. S12). This induction of autophagy might remove damaged mitochondria (mitophagy) and contribute to spermidine induction of mitochondrial OXPHOS in FA-treated cells.

In summary, we discovered an essential role of the DHPS-DOHH-EIF5A[H] pathway in regulating protein synthesis, mitochondrial OXPHOS in hepatic cells. This pathway is dysregulated in FA-treated hepatic cells, a dietary mouse model of NAFLD, and patients with NASH. Spermidine, a substrate for EIF5A hypusination, preserved mitochondrial protein expression, number, and activity via its induction of EIF5A[H] in FA-treated hepatic cells. In mouse model of NASH, spermidine supplementation partially prevented the loss of hepatic EIF5A[H] and mitochondrial protein expression that occurred in NASH, which, in turn, increased FAO and partially prevented hepatosteatosis, inflammation, and fibrosis in NASH. These findings suggest that spermidine, an endogenous polyamine with known anti-aging effects, could have beneficial effects in improving mitochondrial function in NAFLD as well as other metabolic and aging-related diseases.

## Methods

### Gene expression level of Polyamine biosynthesis pathway in patients with NAFLD

For analysis of polyamine metabolism and EIF5A hypusination genes expression in human NASH and steatosis, two publicly available database was used[20,21]. One dataset is accessible at the ArrayExpress public repository for microarray data under the accession number E-MEXP-3291 (http://www.webcitation.org/5zyojNu7T)[20]. The distribution of genes expression in Control ($n = 19$), steatotic ($n = 10$), and NASH ($n = 16$) samples were ascertained via violin plots, and the statistical significance of expression differences across the three groups was determined via one-way ANOVA. All data analysis of using the datasets were performed in the statistical package, R 3.2.3. The other dataset (GSE48452) contains control ($n = 12$), steatosis ($n = 9$), and NASH ($n = 17$)[21].

For analysis of polyamine metabolism and EIF5A hypusination genes expression in patients with mild NAFLD (fibrosis stage 0-1; $n = 35$) and advanced NAFLD (fibrosis stage 3-4, $n = 31$), we retrieved GCRMA normalized signal for genes of interest from microarray dataset (GSE49541) that we have previously reported[47]. Histologic findings were scored using the NASH CRN scoring system[48]. Detailed descriptions of the study population, histologic scoring, RNA

preparation, microarray hybridization, quality control analysis, and data normalization have been reported[47]. Six cases, including normal control ($n = 3$), post-transplantation ($n = 2$), and a case with possible superimposed autoimmune hepatitis ($n = 1$), were excluded from current analysis. Significance was calculated by two-tailed Student's $t$ test or Mann–Whitney $U$ test, as described in Statistical Analysis.

### Animal models

Animal studies that were conducted in accordance with the principles and procedures outlined in the National Institutes of Health Guide for the Care and Use of Laboratory Animals, were approved by the Institutional Animal Care and Use Committee (IACUC) at SingHealth (2015/SHS/1104). The authors confirm that all experiments were performed in accordance with relevant guidelines and regulations. Animals were housed with 40-70% humidity under a 12-hour light/12-hour dark cycle at 21-24 °C, with food and water available ad libitum.

Diet-induced NASH mice were fed with RD Western Diet (D12079B, Research Diets Inc.) supplemented with 15% fructose in drinking water (WDF) as previously described[22]. To determine the effect of NASH on polyamine biosynthesis and Eif5a hypusination, 10-week old male C57BL/6 J mice were fed with normal chow diet (NCD, Specialty Feeds, SF00-100), or WDF for 16 weeks ($n = 8$). To investigate the preventive effect of spermidine in NAFLD progression, ten-week old male C57BL/6 J mice were fed with NCD, WDF, or WDF supplemented with spermidine (WDF + Spd) for 16 weeks (Fig. 5A) ($n = 8$). Spermidine (Sigma, S2626) was supplemented at 3 mM in drinking water. Spermidine stock was prepared as previously described[4]. Briefly, spermidine supplemented drinking water was replaced twice a week, and spermidine freshly added from 1 M aqueous stock (spermidine/HCl pH 7.4), which was kept at −20 °C for no longer than one month. During dissection, blood and liver tissues were collected.

### Biochemical assays

Triglyceride concentrations in the liver homogenate was measured using Triglyceride Colorimetric Assay Kits (10010303; Cayman Chemical Company, Ann Arbor, MI). Hepatic total hydroxyproline content in the livers was measured using Quickzyme Total Collagen assay kit (Quickzyme Biosciences, Leiden, The Netherlands). Mouse serum levels of alanine aminotransferease (ALT), and cholesterol were measured using Alanine Transaminase Activity Assay Kit (ab105134; Abcam), and Cholesterol Assay Kit (ab65390; Abcam). All colorimetric assays were performed according to the manufacturer's protocol.

### Histology

Freshly harvested liver tissues were fixed for 48 h in 10% neutral-buffered formalin, dehydrated, embedded in paraffin and 5 μm sections were stained with hematoxylin and eosin (H&E) or Masson's Trichrome.

### Metabolomics profiling of lipid metabolites in the liver

In accordance with previously established mass spectrometry (MS)-based methods, liver samples were used for quantitative determination of targeted metabolite levels for CEs, triacylglycerols, DAGs, and MAGs by the Metabolomics Core Facility of Duke-NUS Medical School[49]. Liver tissue was homogenized in 50% acetonitrile and 0.3% formic acid. For TAG, DAG, MAG, CE species, the extraction method was adapted from Lipidyzer™ platform protocol (AB Sciex, USA). Briefly, 25 ul of tissue homogenate was solubilised in 25 μL of internal standards (IS kit for Lipidyzer™ platform, AB Sciex, USA), and subjected to biphasic liquid-liquid extraction using a combination of dichloromethane, methanol, and water. Samples were centrifuged at 150 x g for 10 min. The lower phase (organic) layer was transferred to a new tube, dried under nitrogen flow, and reconstituted in methanol for LC-MS analysis.

### Cell culture

Immortalized mouse AML12 hepatic cells were purchased from ATCC (CRL-2254), and maintained at 37 °C in DMEM:F12 Medium containing 10% fetal bovine serum, 10 μg/ml insulin, 5.5 μg/ml transferrin, 5 ng/ml selenium, 10 ng/ml dexamethasone in a 5% $CO_2$ atmosphere.

For siRNA transfection, AML12 cells were trypsinized, mixed with opti-MEM medium (Invitrogen) containing Lipofectamine RNAimax (Invitrogen) and 20 nM DOHH (s97895, Silencer select, Thermo Fisher Scientific), EIF5A (s114771, Silencer select) or control siRNA according to the manufacturer's recommendations.

For FA loading, 0.5% BSA-conjugated FA (0.6 mM palmitic acid, and 0.17 mM oleic acid) were added to cells with or without 100 μM spermidine for 48 h.

For puromycin incorporation assay, 1 μg/ml puromycin was added to culture medium 60 min prior to harvesting protein samples.

### Label-free LC-MS/MS proteomics

The label-free LC-MS/MS proteomics analysis was performed by Novogene (Beijing, China). Pooled ($n = 2$) cell pellets for each experimental condition were lysed in lysis buffer containing 100 mM NH4HCO3(pH 8), 8 M Urea, and 0.2% SDS, followed by 5 min of ultrasonication on ice. The lysate was centrifuged at 12000×$g$ for 15 min at 4 °C. The supernatant was reduced with 10 mM DTT for 1 h at 56 °C, and subsequently alkylated with iodoacetamide for 1 h at room temperature in the dark. Then samples were mixed with four volume of chilled acetone, and incubated at −20 °C for at least 2 h. After centrifugation, the pellet was collected, washed with cold acetone, and dissolved by dissolution buffer [0.1 M triethylammonium bicarbonate (TEAB, pH 8.5), 6 M Urea]. Followed by trypsin digestion, peptide extraction, column separation, the peptides were analyzed by Q Exactive HF-X mass spectrometer (Thermo Fisher), with ion source of Nanospray Flex™ (ESI). The protein sequences were identified using Mus_musculus_uniprot_2021_3_9.fasta (86624 sequences) database using Proteome Discoverer2.2 software. The mass spectrometry proteomics data have been deposited to the ProteomeXchange Consortium via the PRIDE[50] partner repository with the dataset identifier PXD028714. Peptide Spectrum Matches (PSMs) with credibility of >99% was identified as PSMs. The identified protein contains at least 1 unique peptide. The proteins were identified with FDR ≤ 1.0%.

The proteins quantifications were analyzed using the *significance A* test, which was first described by Cox J et al.[51] in the context of SILAC peptide ratios and adapted to our study. Here, to identify proteins that change between our experimental conditions, we implemented an outlier significance score for log protein ratios (*significance A*) for each protein. Briefly, to create a robust (and asymmetrical) estimate of the standard deviation (s.d.) of the main distribution, we first calculated the 15.87, 50, and 84.13 percentiles $r_{-1}$, $r_0$, and $r_1$. $r_1-r_0$ and $r_0-r_{-1}$ are right- and left-sided robust s.d. (In the case of a normal distribution, these would be equal to each other and to the conventional definition of an s.d.). Then, to derive a suitable measure for a protein ratio $r > r_0$ being significantly far away from the main distribution, we used the distance to $r_{0, Z}$ which is measured in terms of the right s.d.

$$z = \frac{r - r_0}{r_1 - r_0} \tag{1}$$

Lastly, to calculate a $P$ value for detection of a significant outlier ratios we define the quantity (2)

$$significance\ A = \frac{1}{2} erfc\left(\frac{1}{\sqrt{2}}\right) = \frac{1}{\sqrt{2\pi}} \int_z^\infty e^{-t^2/2} \mathrm{d}t$$

as the probability of obtaining a protein log-ratio of at least this magnitude under the null hypothesis that the distribution of log-ratios has normal upper and lower tails. The differentially expressed proteins between the experimental conditions were then identified by using two criteria: (1) fold change >1.5-fold and (2) $P$ value <0.05 (obtained by the *significance A* test above).

### Pathway analysis of differentially expressed proteins
Gene Ontology (GO) (GO biological process, and GO cellular component) analysis of differentially expressed proteins were performed by WebGestalt (WEB-based GEne SeT AnaLysis Toolkit, http://www.webgestalt.org/)[52], in which the set of detected proteins in our proteomics analysis was used as the background set for the enrichment (Reference List). A Bonferroni correction was then applied to correct for multiple testing. The venn diagram of overlapped proteins under each pathway was generated at http://bioinformatics.psb.ugent.be/webtools/Venn/.

### RNA purification and RT-PCR
Total cellular RNA was isolated with CatchGene Cell/Tissue RNA kit (CatchGene, MR20250) following the manufacturer's protocol. Total RNA was quantified with a Nanodrop ND-1000 spectrophotometer. Total RNA (1 µg) was reverse-transcribed using iSCRIPT cDNA synthesis kit (Bio-Rad, 1708890) under conditions defined by the supplier. cDNA was quantified by real-time polymerase chain reaction (RT-PCR) on the System (Applied Biosystems HT-7900), SDS software 2.4, version 2.4.1. PCR was performed using QuantiFast SYBR Green PCR Kit (Qiagen) according to the manufacturer's instructions. Sequences of primers were listed in Supplementary Table 2.

### Western blot analysis
Proteins were separated by SDS–PAGE under reducing conditions and transferred to poly(vinylidene fluoride) membranes. Membranes were blocked with 5% nonfat milk in phosphate-buffered saline with 0.1% Tween 20 (Sigma-Aldrich, P9416; PBST). The blots were incubated overnight at 4 °C with primary antibodies. Immunoblot analysis was performed using an enhanced chemiluminescence procedure (GE Healthcare, RPN2106), and western blot images were captured using the ChemiDoc™MP system (Bio-Rad), Image Lab version 6.1.0. Densitometric analysis was performed using ImageJ software (National Institutes of Health). The following antibodies were used: Cell Signaling Technology: anti-COXIV (3E11, #4850), 1:1000; anti-VDAC1 (D73D12, #4661), 1:1000; anti-Tomm20 (D8T4N, #42406), 1:1000; anti-GAPDH (14C10, #2118), 1:10000; anti-Cytochrome c (136F3, #4280), 1:1000; anti-LC3B (#2775), 1:1000; Novus Biologicals: anti-Dhps (NBP1-82648), 1:1000; BD Biosciences: anti-Eif5a (611976), 1:1000; Sigma-Aldrich: anti-Dohh (HPA041953), 1:1000; Millipore: anti-Hypusine (ABS1064), 1:1000; anti-puromycin (12D10, #MABE343), 1:1000; Abcam: anti-Tfam (ab131607), 1:1000; anti-PGC1α (ab54481), 1:1000; anti-TFEB (ab2636), 1:1000; and Atp5a, Uqcrc2, Sdhb, and Ndufb8 were detected using Total OXPHOS Rodent WB Antibody Cocktail (ab110413), 1:1000. Primary antibodies were diluted in 1% BSA containing PBST.

### Cellular oxygen consumption rate analysis
Mitochondrial stress test was performed using Mito stress test kit (Seahorse Biosciences, 103015-100). FAO assay was performed using long chain FAO inhibitor etomoxir (40 µM). Oxygen consumption was measured at 37 °C using an Agilent Seahorse XFe96 Extracellular Flux Analyzer (Seahorse Bioscience Inc., North Billerica, MA, USA), Wave version 2.6.3.5.

For Dohh and Eif5a KD, AML12 cells were transfected with *Dohh, Eif5a* siRNA, or negative control siRNA for 48. For lipid loading, AML12 cells were seeded in 96-well plates, and treated with 0.5% BSA-conjugated FA (0.6 mM palmitic acid, and 0.17 mM oleic acid) with or without 100 µM spermidine for 48 h. For spermidine treatment in *Dohh* KD cells, AML12 cells were transfected with *Dohh*, or negative control siRNA 24 h prior to co-treatment with FA or spermidine for another 48 h.

### BODIPY 493/503 Staining and cellular fat content measurement in vitro
The neutral lipid was stained with fluorescent dye BODIPY 493/503 (5 µg/ml) for 10 min. Subsequently, the cells were washed, resuspended in PBS, and analyzed ($1 \times 10^4$ cells/measurement) using a MACSQuant Analyzer 10 flow cytometer (Miltenyi Biotec), MACSQuantify™ Software version 2.11.5.

### MitoSOX™ staining and mitochondrial ROS measurement
Mitochondrial superoxide was stained with fluorescent dye MitoSOX™ (5 µM) for 10 min at 37 °C, followed by measurement using a MACSQuant Analyzer 10 flow cytometer.

### Mitochondrial DNA content
Total DNA was extracted DNeasy Blood & Tissue Kit (Qiagen). tRNA-Tyr/mt-Col gene was used as marker for mitochondrial DNA content (primers: Fwd: 5′-CAGTCTAATGCTTACTCAGC-3′ Rev: 5′-GGGCAGTTACGATAACATTG-3′), and Lpl gene was used as a marker for nuclear DNA content (Primers: Fwd: 5′-GGATGGACGGTAAGAGTGATTC-3′ Rev: 5′-ATCCAAGGGTAGCAGACAGGT-3′). 20 ng of genomic DNA was used per reaction.

### Statistical analysis
All measurements were taken from distinct samples. In each case, Shapiro–Wilk normality test was performed to assess departure from normality in our data. If the significance value of the Shapiro–Wilk Test was >0.05, we assumed the data were normally distributed and we implemented the parametric two-tailed Student's $t$ test or, when required, One-Way ANOVA followed by Fisher's least significant difference (LSD) test. If the data failed to meet the normality requirement (Shapiro–Wilk Test $P < 0.05$), we used the non-parametric Mann–Whitney $U$ test or Kruskal–Wallis tests. The heatmap of lipidomic profiling was presented as mean value of each group using normalized percentage value for each lipid species, with smallest mean defined as 0%, and largest mean defined as 100%. All the analysis was done using GraphPad Prism version 9.0.1.

## Data availability

The mass spectrometry proteomics data have been deposited to the ProteomeXchange Consortium via the PRIDE[50] partner repository with the dataset identifier PXD028714. This paper analyzes existing publicly available datasets with accession numbers E-MEXP-3291[20], GSE48452[21], and GSE49541[47]. Data generated in this study are provided in the Supplementary Information/Data. Source data are provided with this paper.

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

## Acknowledgements
The authors would like to thank all of the study participants who contributed their biospecimens and data to the Duke University Health System NAFLD Clinical Database and Biorepository and gratefully acknowledge Dr. Manal Abdelmalek, who had led the database project and the research team, including Dr. Cynthia Moylan (Hepatologist) and Dr. Cynthia Guy (Pathologist), referring physicians, surgeons at Duke Center for Metabolic and Weight Loss Surgery, research and data management personnel, study coordinators, and clinical personnel without whom this study would not have been possible. The authors would also like to thank Dr. Tianyi Chen, who helped expression data retrieval for the analysis. The microarray project was supported through an American Recovery and Reinvestment Act (ARRA) grant from the NIAAA: 5RC2 AA019399 (Anna Mae Diehl, Principal Investigator). This work was supported by the Ministry of Health, A*STAR, and National Medical Research Council Singapore grants MOH-000306 (MOH-CSA-SI19may-0001) to PMY; NMRC/OFYIRG/0002/2016 and MOH-000319 (MOH-OFIRG19may-0002) to BKS; NMRC/OFYIRG/077/2018 to MT. Duke-NUS Medical School and Estate of Tan Sri Khoo Teck Puat Khoo Pilot Award (Collaborative) Duke-NUS-KP(Coll)/2018/0007 A to J.Z.

## Author contributions
J.Z., J.P., and P.M.Y. conceived the experiments; J.Z., J.P., M.T., J.P.H., A.W., S.G.S., and B.K.S. performed the experiments; J.Z., J.P., B.K.S., and A.W. analyzed the data; A.W. and S.A.C. established the protocol; E.P. supervised pathway analysis of proteomics data; A.S., A.M.D. provided and analyzed microarray dataset GSE49541; J.Z. and P.M.Y. wrote the article. All authors provided critical feedback, and helped shape the research, analysis, and article.

## Competing interests
The authors declare no competing interests.

## Additional information

[1]Program of Cardiovascular & Metabolic Disorders, Duke-NUS Medical School Singapore, 8 College Road, Singapore 169857, Singapore. [2]Medical Research Council, London Institute for Medical Sciences, Imperial College London, London, UK. [3]National Heart Centre, Singapore, Singapore. [4]Division of Gastroenterology, Duke University School of Medicine, Durham, NC, USA. [5]Institute for Big Data and Artificial Intelligence in Medicine, School of Science, China Pharmaceutical University, Nanjing, China. [6]Duke Molecular Physiology Institute and Sarah W. Stedman Nutrition and Metabolism Center, Durham, NC, USA. [7]Duke University School of Medicine, Durham, NC, USA. [8]These authors contributed equally: Jin Zhou, Jeremy Pang. ✉e-mail: jin.zhou@duke-nus.edu.sg; paul.yen@duke-nus.edu.sg

