## [Peer Review File · Nature Communications]

Title: Spermidine-mediated hypusination of translation factor EIF5A improves mitochondrial fatty acid oxidation and prevents NASH progressionREVIEWER COMMENTS

Reviewer #1 (Remarks to the Author):

In this interesting study, Zhou, Pang, and colleagues examine how fatty acids modulate mitochondrial oxidation, and more interestingly, how the naturally-occurring polyamine spermidine affects EIF5AH expression, downstream mitochondrial protein expression, mitochondrial oxidation and NASH progression through deoxyhypusine synthase (DHPS) and deoxyhypusine hydroxylase (DOHH) expression in vitro and in vivo. They also show human data demonstrating that DHPS and DOHH mRNA expression is reduced in humans with NASH. The authors appropriately conclude that the DHPS/DOHH-EIF5A-fatty acid oxidation pathway is an attractive target for NAFLD/NASH. There is no doubt that the topic is translationally important (as shown by the authors' own data) and the studies are generally well performed. I do have several concerns, which I would rate as minor-to-moderate, but I am supportive of the work in general and believe that with appropriate revisions it will be of great interest to the readers of Nature Communications.

1. In my view, the implication of the novelty of the observation that fatty acids impair mitochondrial health is overstated; this has been examined many times (please see PMID 26589966 and references therein) and incorporate appropriate discussion of the literature on this topic.
2. The OCR data with etomoxir (Fig 2e, 4f) are concerning because of the precipitous drop in OCR even without etomoxir. These OCR rates are expected to be far more stable, as shown in Fig 2d. Please repeat or omit (I think the latter would be fine; the ATP formation is much more compelling anyway).
3. I am concerned that there is a technical problem with the triglyceride assay. Triglyceride concentrations are generally 5-10x higher in all groups, although the relative differences between control and WDF are reasonable. (See PMID 15591160, 30117227, 27572941, and many others.) I would request that the authors double check these measurements and provide raw data (absorbance, standard curve, ideally photo of the spectrophotometric plate etc) in the rebuttal.

Reviewer #2 (Remarks to the Author):

Zhou et al. showed reduced EIF5A hypusination in cell culture subjected to lipid accumulation and mouse models that mimic NAFLD. This led to reduced global protein production, which affected particularly mitochondrial proteins, causing reduced mitochondrial activity and fatty acid beta-oxidation. Then the authors went a step further showing spermidine treatment partially (in many cases nearly completely) rescued the reduced protein production defects and mitochondrial function and even prevented NAFLD progression in the mice itself. Although I am not an expert on mitochondrial function and on NAFLD (and consequently NASH), I do find the presented results very interesting and potentially very impactful. However, several critical things need to be addressed before this manuscript could be potentially accepted.

Major points

In general, the statistical rigor of the proteomics measurements is rather weak. First of all, essential information about the experiments themselves and the subsequent analysis is not provided (see below for more details). However, having said that, the authors are indeed careful about the conclusions drawn from the proteomics experiments, which makes their conclusions in general robust. The authors never focus on protein changes of single genes but only “metanalysis” of gene groups (by GO term analysis). For such “trend analysis” of gene groups, the data presented should be good enough, given that all the information below is provided.

1. Not enough information is provided how the proteomics experiments have been conducted.

- How many replicates (what is the n)?
- Any kind of statistical analysis or just fold changes?
- How many proteins identified/quantified in each experiment in total? (that info is provided only for the first experiment (Figure 2), but not the subsequent proteomics experiments)
- How did the proteomics measured changes agree with the western blot determined protein changes? My guess here is that the protein quantification was not reliable enough due to the lack of replicates, but if that is the case, this should be clearly stated and also the reason why the analysis is only at the group level, in this case mainly GO terms.

2. Also there is not enough information about the GO term analysis provided

- What was the background set for the analysis – all measured proteins or the whole predicted proteome? That is important – it should be only the measured proteins to correct for measurement bias.
- Did they also try as an input simply a ranked list (based on fold changes)? Do they get similar results?
- The GO term associated p-values – are they corrected p-values or only p-values? They should be corrected and if they were – what was the correction that was applied?
- GO term analysis for upregulated as well as down-regulated should be given. The less important ones can be moved to supplementary information, but both should be provided in order to provide the full picture and not make the impression of “cherry picking”

3. Western blots: a lot of their conclusions are based on protein quantifications via Western blots.

Therefore, it is important to be as thorough and precise about the quantifications as possible. However, I do not think that normalizing total protein loading by calculating the ratio to one protein only is the accepted standard nowadays. This is especially true if a metabolic enzyme like GAPDH is used in this case. The authors are actually claiming throughout the manuscript that metabolism is affected (granted they focus on mitochondrial function and not glycolysis, but metabolic pathways are in general interconnected). Normally the protein loading should be normalized by integrating nearly the whole lane intensity after Coomassie (or even Ponceau S) staining. To use only one protein as loading control, like the authors did, risks that this protein itself also shows potential expression changes between conditions. Again – especially in a paper like the one presented where so many conclusions depend on protein quantifications via Western blotting a more careful and robust protein normalization procedure should be chosen.

4. Figure 2A: how well did the Dohh knockdown really work in the siDohh experiment? No quantification of the Dohh levels is provided, but it seems from the blots that the Dohh knockdown by siRNAs is rather weak – e.g. much weaker than in the eIF5A knockdown experiment where Dohh levels seem stronger affected. Could the authors add that quantification information?

Minor points

5. “Protein translation” is not a correct scientific term. The authors use it throughout the manuscript. It should be either “mRNA translation” or “protein production/synthesis”. Please change that throughout the manuscript.

6. The authors consistently conclude throughout the manuscript that the fact that protein levels change, but often not mRNA levels, proves that the change in expression is due to change in translation rates. Although this is by far the most likely explanation due to the nature of eIF5A as an elongation factor, the authors did not definitely show that. Potentially other post-transcriptional mechanism cannot be 100% excluded, such as changes in protein stability instead (or in addition to) of translation rate changes. In my opinion, the authors should be cautious about the wording and also clearly state that this at this stage is only a hypothesis (even if a very likely one).

Reviewer #3 (Remarks to the Author):

The work by Yen and colleagues is highly interesting, since it provides an option for NASH treatment, a disease that affects one quarter of the elderly population. In addition, the experiments are technically well done, but leave some room for improvements.

Major:

- It has been demonstrated by a couple of labs that Spermidine mediated hypusination is causing autophagy (and/or mitophagy). So, it might well be that improved mitophagy plays a role in some of the observed effects (e.g. enhanced respiration). This should be tested in a different set of samples/assays, e.g. by evaluating the localization of TFEB and blotting LC3.

- Fig. 1b and 1c: These are very interesting data, but definitely need support from an independent and targeted measurement in patients with steatosis and NASH. The changes are rather small and, given (i) their importance for the story and (ii) different outcomes, they should be independently validated in a different cohort, rather than relying solely on publicly available databases.

- General: eIF5A-hypusine levels should and must always be (additionally) normalized to individual

eIF5A-total levels

For instance, Fig. 2a: it appears that ieIF5A works much better at reducing hypusine levels, than siDohh. However, when normalization is performed on total eIF5A levels, the analysis might look different.

- Cell culture: When adding spermidine to FBS-containing media it is crucial to control for indirect effects of spermidine-degradation products by amine oxidases. There are a couple of papers and methods published on this topic and the authors should repeat at least the key cell culture experiments with the respective controls. Otherwise, some of the effects observed for spermidine-supplementation could be unspecific and unrelated to the polyamine's biological functions.

- Statistics:

o The authors do not state whether the data were checked for normality. At least some of the data sets seem to be distributed in a non-normal manner. Hence, the applied tests are likely incorrect and may lead to false conclusions. The authors should re-check all the data in the manuscript and report which test was applied for which set of data.

Minor:

- Line 93: "... spermidine was able to preserve eIF5AH in NAFLD..." The authors mean "in NAFLD mouse model", right? Please specify

- Can the authors elucidate on whether the ratio of hypusinated eIF5A to non-hypusinated eIF5A is important or rather the overall level of eIF5A-Hypusination?

- Some findings are overstated: E.g. Line 362 "Remarkably, spermidine supplementation reversed ALL of these effects." This is partly true, but the extent of rescue is for some effects only little. Hence, rephrasing such statements throughout the manuscript to reflect the PARTIAL rescue of spermidine on the alterations in NAFLD/NASH is needed.

- P-values: Please indicate p-value figures, instead of asterisks

- Please provide information on which antibodies were used at which dilutions etc.

Response to Reviewer Comments

Reviewer #1 (Remarks to the Author):

In this interesting study, Zhou, Pang, and colleagues examine how fatty acids modulate mitochondrial oxidation, and more interestingly, how the naturally-occurring polyamine spermidine affects EIF5AH expression, downstream mitochondrial protein expression, mitochondrial oxidation and NASH progression through deoxyhypusine synthase (DHPS) and deoxyhypusine hydroxylase (DOHH) expression in vitro and in vivo. They also show human data demonstrating that DHPS and DOHH mRNA expression is reduced in humans with NASH. The authors appropriately conclude that the DHPS/DOHH-EIF5A-fatty acid oxidation pathway is an attractive target for NAFLD/NASH. There is no doubt that the topic is translationally important (as shown by the authors' own data) and the studies are generally well performed. I do have several concerns, which I would rate as minor-to-moderate, but I am supportive of the work in general and believe that with appropriate revisions it will be of great interest to the readers of Nature Communications.

1. *In my view, the implication of the novelty of the observation that fatty acids impair mitochondrial health is overstated; this has been examined many times (please see PMID 26589966 and references therein) and incorporate appropriate discussion of the literature on this topic.*

Response: We appreciate the Reviewer's comments. We revised the Discussion and have incorporated references from the literature reporting fatty acid-induced dysfunctional mitochondria in hepatic cells and NAFLD.

2. *The OCR data with etomoxir (Fig 2e, 4f) are concerning because of the precipitous drop in OCR even without etomoxir. These OCR rates are expected to be far more stable, as shown in Fig 2d. Please repeat or omit (I think the latter would be fine; the ATP formation is much more compelling anyway).*

Response: We appreciate the Reviewer's comments. We removed Fig 2e and 4f as per Reviewer's comments.

3. *I am concerned that there is a technical problem with the triglyceride assay. Triglyceride concentrations are generally 5-10x higher in all groups, although the relative differences between control and WDF are reasonable. (See PMID 15591160, 30117227, 27572941, and many others.) I would request that the authors double check these measurements and provide raw data (absorbance, standard curve, ideally photo of the spectrophotometric plate etc) in therebuttal.*

Response: We appreciate the Reviewer's comments. We rechecked the raw data and found that although the R^2 of standard curve was 0.9991, the blank well showed an absorbance at 0.25, which ideally should be about 0.05. Thus, we re-performed this assay, the trend was similar as before, but the TG content was 5-10 times higher than the previous values (Fig.1 below). We thank the Reviewer for pointing out the discrepant low values which prompted us to remeasure the triglyceride levels. We have now shown these new results in new Figure 5c.

Figure.1 Hepatic triglyceride content in mice fed on normal chow (control), WDF, or WDF+spermidine (WDF+Spd).

Reviewer #2 (Remarks to the Author):

Zhou et al. showed reduced eIF5A hypusination in cell culture subjected to lipid accumulation and mouse models that mimic NAFDL. This lead to reduced global protein production, which affected particularly mitochondrial proteins, causing reduced mitochondrial activity and fatty acid beta-oxidation. Then the authors went a step further showing spermidine treatment partially (in in many cases nearly completely) rescued the reduced protein production defects and mitochondrial function and even prevented NAFDL progression in the mice itself. Although I am not an expert on mitochondrial function and on NAFDL (and consequently NASH), I do find the presented results very interesting and potentially very impactful. However, several critical things need to be addressed before this manuscript could bepotentially accepted.

Major points

In general, the statistical rigor of the proteomics measurements is rather weak. First of all, essential information about the experiments themselves and the subsequent analysis is not provided (see below for more details). However, having said, the authors are indeed careful about the conclusions drawn from the proteomics experiments, which makes their conclusions in general robust. The authors never focus on protein changes of single genes but only “metanalysis” of gene groups (by GO term analysis). For such “trend analysis” of gene groups, the data presented should be good enough, given that all the information below is provided.

1. *Not enough information is provided how the proteomics experiments have been conducted.*

- *How many replicates (what is the n)?*

Response: The proteomics experiments were performed using pooled (n=2) cell pellets for each experimental condition. Although we only obtained one reading for each condition, we believe that pooling the duplicates was helpful in reducing false discovery. We have now provided this information in “Material and Methods”.

- *Any kind of statistical analysis or just fold changes?*

Response: We apologize that we did not mention the statistical analysis regarding the protein quantification. The protein quantifications were analysed using the *significance A* test, which was first described by Cox J *et al*¹ in the context of SILAC peptide ratios and adapted to our study. Here, to identify proteins that change between our experimental conditions, we implemented an “outlier significance score” for log protein ratios (*significance A*) for each protein. Briefly, to create a robust (and asymmetrical) estimate of the standard deviation (s.d.)

of the main distribution, we first calculated the 15.87, 50 and 84.13 percentiles r_{-1} , r_0 , and r_1 . $r_1 - r_0$ and $r_0 - r_{-1}$ are right- and left-sided robust s.d. (In the case of a normal distribution, these would be equal to each other and to the conventional definition of an s.d.). Then, to derive a suitable measure for a protein ratio $r > r_0$ being significantly far away from the main distribution, we used the distance to r_0 , z , which is measured in terms of the right s.d.

$$z = \frac{r - r_0}{r_1 - r_0}$$

Lastly, to calculate a P-value for detection of a significant outlier ratios we define the quantity

$$\text{significance } A = \frac{1}{2} \operatorname{erfc} \left(\frac{1}{\sqrt{2}} \right) = \frac{1}{\sqrt{2\pi}} \int_z^\infty e^{-t^2/2} dt$$

as the probability of obtaining a protein log-ratio of at least this magnitude under the null hypothesis that the distribution of log-ratios has normal upper and lower tails. The differentially expressed proteins between the experimental conditions were then identified by using two criteria: (1) fold change > 1.5 fold and (2) P-value < 0.05 (obtained by the *significance A* test above). We have now included this information in “Material and Methods”.

- *How many proteins identified/quantified in each experiment in total? (that info is provided only for the first experiment (Figure 2), but not the subsequent proteomics experiments)*

Response: We have now provided this information as New Supplementary Table 1.

Supplementary Table 1. Summary of differentially expressed proteins

Compared Samples	Number of proteins	regulated Type	fold-change > 1.5
siDohh vs Control	4286	up-regulated	297
		down-regulated	290
FA vs Control	4272	up-regulated	254
		down-regulated	332
FA+Spd vs FA	4266	up-regulated	326
		down-regulated	361
FA+Spd+siDohh vs FA+Spd	4271	up-regulated	270
		down-regulated	301

- *How did the proteomics measured changes agree with the western blot determined protein changes? My guess here is that the protein quantification was not reliable enough due to the lack of replicates, but if that is the case, this should be clearly stated and also the reasoned why the analysis is only at the group level, in this case mainly GO terms.*

Response: As the Reviewer pointed out, due to the lack of replicates from proteomics data, we were not able to compare the quantification of each individual protein, and only analyzed the data at the group level by performing GO pathway analysis. We have now stated this point in the “Results”.

2. *Also there is not enough information about the GO term analysis provided*

- *What was the background set for the analysis – all measured proteins or the whole predicted proteome? That is important – it should be only the measured proteins to correct for measurement bias.*

Response: We appreciate Reviewer’s comments. For the initial submission, the background set for the analysis was the predicted proteome. We have now re-performed the GO term analysis by WebGestalt (WEB-based GENE SeT AnaLysis Toolkit, <http://www.webgestalt.org/>)², using measured proteins as the background set. We now have included this information in “Material and Methods”

- *Did they also try as an input simply a ranked list (based on fold changes)? Do they get similar results?*

Response: We appreciate Reviewer’s comments. We performed GO term analysis using ranked list at WebGestalt (WEB-based GENE SeT AnaLysis Toolkit, <http://www.webgestalt.org/>)². We have found similar results that down-regulated pathways in Dohh KD cells were enriched in mitochondrial and ribosomal proteins. These pathways were down-regulated in FA-treated cells, and were up-regulated by spermidine treatment. The results were summarized in the table below.

		GO term	P value
siDohh vs Control	Down-regulated	GO:0005759 mitochondrial matrix	0
		GO:0005840 Ribosome	0
		GO:0070069 cytochrome complex	0.03
		GO:0006413 translational initiation	0.04
		GO:0072350 tricarboxylic acid metabolic process	0.08
FA vs control	Down-regulated	GO:0044455 mitochondrial membrane part	0
		GO:0030964 NADH dehydrogenase complex	0
		GO:0045120 pronucleus	0
		GO:0030684 preribosome	0.01
	Up-regulated	GO:0005811 lipid droplet	0.04
FA+Spd Vs FA	Up-regulated	GO:0030684 preribosome	0.02
		GO:0022613 ribonucleoprotein complex biogenesis	0.03
		GO:0140053 mitochondrial gene expression	0.04
		GO:0044455 mitochondrial membrane part	0.08
FA+Spd+si Dohh vs FA+Spd	Down-regulated	GO:0072350 tricarboxylic acid metabolic process	0.04

- *The GO term associated p-values – are they corrected p-values or only p-values? They should be corrected and if they were – what was the correction that was applied?*

Response: We appreciate Reviewer’s comments. We re-performed the pathway analysis, and have shown corrected p-value in the revised manuscript. Bonferroni correction was applied for multiple-comparison correction. We have now included this information in the “Material and Methods” and “Figure Legends”.

GO term analysis for upregulated as well as down-regulated should be given. The less important ones can be moved to supplementary information, but both should be provided in order to provide the full picture and not make the impression of “cherry picking”

Response: We appreciate Reviewer’s comments. We have now included the GO term analysis for both upregulated and downregulated proteins with the corrected p-values in the revised manuscript (New Figs. S5, 8 and 11).

3. *Western blots: a lot of their conclusions are based on protein quantifications via Western blots. Therefore, it is important to be as thorough and precise about the quantifications as possible. However, I do not think that normalizing total protein loading by calculating the ratio to one protein only is the accepted standard nowadays. This is especially true if a metabolic enzyme like GAPDH is used in this case. The authors are actually claiming throughout the manuscript that metabolism is affected (granted they focus on mitochondrial function and not glycolysis, but metabolic pathways are in general interconnected). Normally the protein loading should be normalized by integrating nearly the whole lane intensity after Coomassie (or even Ponceau S) staining. To use only one protein as loading control, like the authors did, risks that this protein itself also shows potential expression changes between conditions. Again – especially in a paper like the one presented where so many conclusions depend on protein quantifications via Western blotting a more careful and robust protein normalization procedure should be chosen.*

Response: We strongly agree with Reviewer that a reliable quantification of Western blots is important. We always perform Ponceau S staining after transfer to make sure that the transfer is well performed, and the staining is generally uniform across the various conditions which indicates approximately equal loading of all samples on the gel before electrophoresis. We also found that GAPDH protein level remained unchanged under our experimental conditions although we don’t know if its activity was affected. Accordingly, we used GAPDH as internal control for normalization. We have now included images of Ponceau S staining for each condition in New Figs. S2, 3, 6, 9, 10 and 14 for the Reviewer.

4. *Figure 2A: how well did the Dohh knockdown really work in the siDohh experiment? No quantification of the Dohh levels is provided, but it seems from the blots that the Dohh knockdown by siRNAs is rather weak – e.g. much weaker than in the eIF5A knockdown experiment where Dohh levels seem stronger affected. Could the authors add that quantification information?*

Response: We apologize that we did not present a better representative Dohh blot during the initial submission. Knockdown of Dohh resulted in robust decreases in Dohh mRNA (Fig. 1a) and protein levels (Figure 1b below). We now show an improved representative Dohh blot with densitometry analysis in new Fig. 2a, and have included the mRNA expression of Dohh after knockdown in new Fig. S4.

Figure 1. Dohh mRNA (a) and protein (b) expression. AML12 cells were transfected with 20 nM of negative (siNeg), Dohh (siDohh) or eIF5A (siEIF5A) siRNA for 48 h.

Minor points

5. “Protein translation” is not a correct scientific term. The authors use it throughout the manuscript. It should be either “mRNA translation” or “protein production/synthesis”. Please change that throughout the manuscript.

Response: We appreciate Reviewer’s comments. We have now used the term “protein synthesis” in the revised manuscript.

6. The authors consistently conclude throughout the manuscript that the fact that protein levels change, but often not mRNA levels, proves that the change in expression is due to change in translation rates. Although this is by far the most likely explanation due to the nature of eIF5A as an elongation factor, the authors did not definitely show that. Potentially other post-transcriptional mechanism cannot be 100% excluded, such as changes in protein stability instead (or in addition to) of translation rate changes. In my opinion, the authors should be cautious about the wording and also clearly state that this at this stage is only a hypothesis (even if a very likely one).

Response: We appreciate Reviewer’s comments. We have now included this point in the Discussion.

Reviewer #3 (Remarks to the Author):

The work by Yen and colleagues is highly interesting, since it provides an option for NASH treatment, a disease that affects one quarter of the elderly population. In addition, the experiments are technically well done, but leave some room for improvements.

Major:

- It has been demonstrated by a couple of labs that Spermidine mediated hypusination is causing autophagy (and/or mitophagy). So, it might well be that improved mitophagy plays a role in some of the observed effects (e.g. enhanced respiration). This should be tested in a different of samples/assays, e.g. by evaluating the localization of TFEB and blotting LC3.

Response: We appreciate Reviewer’s comments. We assessed autophagy status, and found that FA treatment reduced the level of TFEB and LC3B-II, which was restored by spermidine. Of note, in Dohh KD cells, spermidine failed to restore the levels of TEFB and LC3B-II (Figure 2 below). These results indicated that spermidine restored autophagy in FA-treated cells in an eIF5A^H-dependent manner, which might be another mechanism involved in spermidine regulation of mitochondrial activity. We have now included this point in the Results (new Fig. S12) and Discussion.

Figure 2. Spermidine restored autophagy in FA-treated cell in EIF5A^H-dependent manner. Western blot and densitometric analysis of TFEB and LC3B-II. AML12 cells were first transfected with 20 nM of negative (siNeg), or Dohh (siDohh) siRNA for 24 h, followed by treatment with BSA-conjugated FA (palmitic acid 0.6 mM, oleic acid 0.17 mM) with or without spermidine (100 μM) for 48 h.

- Fig. 1b and 1c: These are very interesting data, but definitely need support from an independent and targeted measurement in patients with steatosis and NASH. The changes are rather small and, given (i) their importance for the story and (ii) different outcomes, they should be independently validated in a different cohort, rather than relying solely on publicly available databases.

Response: We thank the Reviewer for pointing out the different outcomes which prompted us to re-investigate the datasets and our analysis. Although we have not found substantial evidence to explain the different outcome between the two datasets, we did find that one dataset (GSE48452) contains a few samples which was collected post-Bariatric Surgery. We now have excluded these samples and re-analyzed dataset GSE48452 (new Fig. 1c).

To validate our findings, we first collaborated with Prof. Pierce Chow (National Cancer Center Singapore) and performed q-PCR analysis of mRNA expression of enzymes in polyamine biosynthesis and EIF5A hypusination pathways in patients with liver cancer that underwent surgical procedure. The liver tissue used for the analysis were harvested from sites distant from the tumor and exhibited no visible evidence of tumor upon surgical assessment. The Control (n=4), Steatosis (n=5) and NASH (n=4) patients were further identified by histological analysis of their liver tissue. The mRNA levels of polyamine biosynthesis enzymes did not change in NASH while *DOHH* mRNA was down-regulated in patients with steatosis and NASH (Fig. 3 below). However, we did not include them in the manuscript due to the relatively small number of samples analyzed.

To further validate our finding in a different cohort, we collaborated with Prof. Anna Mae Diehl (Duke University School of Medicine) and retrieved expression data of genes of interest from microarray dataset (GSE49541) that Diehl's laboratory previously reported³. Due to limited access to normal controls, only patients with "mild" NAFLD (fibrosis stage 0-1; n=35) and "severe" NAFLD (fibrosis stage 3-4, n=31) were included in the current analysis. We found that patients with severe NAFLD showed decreased ARG1 and increased SMS mRNA levels, which may result in decreased spermidine synthesis and increased conversion of spermidine to spermine. DHPS also was decreased in patients with severe NAFLD compared to patients with mild NAFLD (Figure 4 below). All these observed changes may lead to decreased EIF5A^H in patients with severe NAFLD and support findings from the two other cohorts and our in vivo and cell culture experiments. We have included this data as new Fig. S1.

Figure 3. Quantitative-PCR analysis of mRNA expression of polyamine biosynthesis and EIF5A hypusination enzymes in Control (n=4), Steatosis (n=5), and NASH (n=4) patients.

Figure 4. Violin plots showing mRNA levels of genes involved in endogenous polyamine biosynthesis and EIF5A hypusination in patients with “mild” NAFLD (fibrosis stage 0-1; n=35) and “severe” NAFLD (fibrosis stage 3-4, n=31). GCRMA normalized signal was retrieved from microarray dataset (GSE49541)³.

- General: *eIF5A*-hypusine levels should and must always be (additionally) normalized to individual *eIF5A*-total levels

For instance, Fig. 2a: it appears that *siEIF5A* works much better at reducing hypusine levels, than *siDohh*. However, when normalization is performed on total *eIF5A* levels, the analysis might look different.

Response: We appreciate Reviewer’s comments. We have now included the $eIF5A^H/eIF5A$ ratio in all relevant densitometric analysis (new Fig. 1e, 1h, 2a, 3a, 4a, 6c).

-Cell culture: When adding spermidine to FBS-containing media it is crucial to control for indirect effects of spermidine-degradation products by amine oxidases. There are a couple of papers and methods published on this topic and the authors should repeat at least the key cell culture experiments with the respective controls. Otherwise, some of the effects observed for spermidine-supplementation could be unspecific and unrelated to the polyamine’s biological functions.

Response: We appreciate Reviewer’s comments. To rule out the possibility that the observed effects of spermidine on mitochondrial proteins were due to oxidized spermidine products from amine oxidases in the FBS, we cultured AML12 cells using serum free medium, and treated cells with no FA (control), FA, and FA+Spd. We have found that in serum free medium, spermidine supplementation was able to restore levels of $eIF5A^H$ and mitochondrial proteins (New Fig. S9). Together with our finding that spermidine failed to increase mitochondrial proteins in Dohh KD cells (Figure 4b), we believe that the effect of spermidine on mitochondrial protein synthesis depends on its effect on EIF5A hypusination.

- Statistics:

The authors do not state whether the data were checked for normality. At least some of the data sets seem to be distributed in a non-normal manner. Hence, the applied tests are likely incorrect and may lead to false conclusions. The authors should re-check all the data in the manuscript and report which test was applied for which set of data.

Response: We appreciate Reviewer’s comments. We have now re-performed statistical analysis. We performed Shapiro-Wilk normality test for all the data. If the data passed normality test, two-tailed Student’s t-test or One-Way ANOVA analysis was performed. If the

data failed the normality test, Mann-Whitney U-test or Kruskal-Wallis tests were performed. We now have specified the statistical analysis in the Methods and Figure legends.

Minor:

- Line 93: "... spermidine was able to preserve eIF5A^H in NAFLD..." The authors mean "in NAFLD mouse model", right? Please specify

Response: We appreciate Reviewer's comments. We were referring to our mouse model of NAFLD. We now have revised the manuscript accordingly.

- Can the authors elucidate on whether the ratio of hypusinated eIF5A to non-hypusinated eIF5A is important or rather the overall level of eIF5A-Hypusination?

Response: We appreciate Reviewer's comments. eIF5A^H/eIF5A ratio might be a very good indication for DHPS-DOHH expression and/or activity. In the current study, we observed a decrease in both eIF5A^H and eIF5A^H/eIF5A ratio when DOHH was decreased. However, the eIF5A^H/eIF5A ratio also can be affected by the total eIF5A level. If there were similar decreases in both eIF5A^H and eIF5A levels, the ratio might be unchanged. Thus, we feel that eIF5A^H and eIF5A levels, and the eIF5A^H/eIF5A ratio should be determined and presented to assess this pathway.

- Some findings are overstated: E.g. Line 362 "Remarkably, spermidine supplementation reversed ALL of these effects." This is partly true, but the extent of rescue is for some effects only little. Hence, rephrasing such statements throughout the manuscript to reflect the PARTIAL rescue of spermidine on the alterations in NAFLD/NASH is needed.

Response: We appreciate Reviewer's comments. We have now revised the manuscript to reflect the partial rescue of spermidine.

- P-values: Please indicate p-value figures, instead of asterisks

Response: We have now shown p-values instead of using asterisks as per Reviewer's comments.

- Please provide information on which antibodies were used at which dilutions etc.

Response: We have now included the antibodies' information in Material and Methods.

References

1. Cox J, Mann M. MaxQuant enables high peptide identification rates, individualized p.p.b.-range mass accuracies and proteome-wide protein quantification. *Nat Biotechnol* **26**, 1367-1372 (2008).
2. Liao Y, Wang J, Jaehnig EJ, Shi Z, Zhang B. WebGestalt 2019: gene set analysis toolkit with revamped UIs and APIs. *Nucleic Acids Res* **47**, W199-W205 (2019).
3. Moylan CA, *et al.* Hepatic gene expression profiles differentiate presymptomatic patients with mild versus severe nonalcoholic fatty liver disease. *Hepatology* **59**, 471-482 (2014).

REVIEWERS' COMMENTS

Reviewer #1 (Remarks to the Author):

The authors have responded well to my comments. I am pleased that they repeated the triglyceride assay and generated data that make much more sense. They are to be commended on this interesting study.

Reviewer #2 (Remarks to the Author):

The authors have addressed nearly all of my concerns and in general I do support publication. There is only one minor point that I listed below. I do think the authors should address this before publication.

I originally voiced concerns that the authors only use GAPDH to normalize their Western blot results and suggested to use global loading (measured either via Coomassie or Ponceau S) as a normalization factor. I do appreciate that the authors added all the relevant PonceauS stainings. By eye the loading indeed looks very even, but it is hard to put a quantitative value to it as currently presented. Moreover, the authors do claim that GAPDH expression is unchanged under all their conditions and therefore argue that they will stick to using GAPDH as their normalization factor. Maybe I missed that data, but how did the authors determine that GAPDH is not changing? Did they look if GAPDH levels closely match the PonceauS signal or what method was applied? If the authors do want to stick to GAPDH based normalization, showing that GAPDH is really not changing is important. I know that it might seem that I am nitpicking here, but as so many of the conclusion depend on rather small protein level changes detected by western blotting, correct normalization is of utter importance.

Reviewer #3 (Remarks to the Author):

Good revision

Reviewer #2 (Remarks to the Author):

The authors have addressed nearly all of my concerns and in general I do support publication. There is only one minor point that I listed below. I do think the authors should address this before publication.

I originally voiced concerns that the authors only use GAPDH to normalize their Western blot results and suggested to use global loading (measured either via Coomassie or Ponceau S) as a normalization factor. I do appreciate that the authors added all the relevant PonceauS stainings. By eye the loading indeed looks very even, but it is hard to put a quantitative value to it as currently presented. Moreover, the authors do claim that GAPDH expression is unchanged under all their conditions and therefore argue that they will stick to using GAPDH as their normalization factor. Maybe I missed that data, but how did the authors determine that GAPDH is not changing? Did they look if GAPDH levels closely match the PonceauS signal or what method was applied? If the authors do want to stick to GAPDH based normalization, showing that GAPDH is really not changing is important. I know that it might seem that I am nitpicking here, but as so many of the conclusion depend on rather small protein level changes detected by western blotting, correct normalization is of utter importance.

Response: We appreciate Reviewer's comments. We now have included Ponceau S staining together with GAPDH blot and density in the revised manuscript (new Supplementary Figures 2, 3, 6, 9, 10, 14). The analysis showed that there is no statically significant difference of GAPDH density between the experimental conditions.